# Augmenting Online Algorithms
# with $\varepsilon$-Accurate Predictions

**Anupam Gupta**
Carnegie Mellon University
anupamg@cs.cmu.edu

**Debmalya Panigrahi**
Duke University
debmalya@cs.duke.edu

**Bernardo Subercaseaux**
Carnegie Mellon University
bsuberca@cs.cmu.edu

**Kevin Sun**
Duke University
ksun@cs.duke.edu

## Abstract

A growing body of work in learning-augmented online algorithms studies how online algorithms can be improved when given access to ML predictions about the future. Motivated by ML models that give a confidence parameter for their predictions, we study online algorithms with predictions that are $\varepsilon$-accurate: namely, each prediction is correct with probability (at least) $\varepsilon$, but can be arbitrarily inaccurate with the remaining probability. We show that even with predictions that are accurate with a small probability and arbitrarily inaccurate otherwise, we can dramatically outperform worst-case bounds for a range of classical online problems including caching, online set cover, and online facility location. Our main results are an $O(\log(1/\varepsilon))$-competitive algorithm for caching, and a simple $O(1/\varepsilon)$-competitive algorithm for a large family of covering problems, including set cover, network design, and facility location, with $\varepsilon$-accurate predictions.

## 1 Introduction

The study of online algorithms with ML predictions has gained significant traction in the ML and algorithms research communities in recent years. The basic premise is that by having access to predictions about the future, one can design online algorithms that significantly outperform worst-case bounds. In practice, ML models producing such predictions often come with a *confidence* estimate which can be roughly interpreted as a probability $\varepsilon$ of the prediction being accurate—we call these *$\varepsilon$-accurate* predictions. Thus, it is natural to ask: *can we design online algorithms whose performance is a function of the confidence parameter $\varepsilon$ of ML predictions?* If $\varepsilon = 1$ and all predictions are correct, we would like the algorithm to approach optimal offline performance (*consistency*). If $\varepsilon = 0$, then the predictions have no guarantees and we would like to ensure that the algorithm is not much worse than an online algorithm without predictions (*robustness*). Between these two extremes, we would like the performance of the algorithm to *gracefully degrade* with the value of $\varepsilon$. In this paper, we present online algorithms augmented with $\varepsilon$-accurate predictions for fundamental problems such as caching, online set cover, online facility location, and online network design.

**$\varepsilon$-accurate predictions.** We consider ML predictions that are (independently) correct with probability at least $\varepsilon$. Note that $\varepsilon$ represents only a *minimum guarantee* on the probability of a prediction being accurate; individual predictions are allowed to have higher confidence values. Moreover, we do not restrict what an incorrect prediction constitutes since this will depend on specific problem settings. But morally, a correct prediction reveals information about the future while an incorrect prediction does not. The design of online algorithms augmented with $\varepsilon$-accurate predictions has two main challenges: (i) that *information (correct predictions) is provided to the algorithm at a slow rate*, about

36th Conference on Neural Information Processing Systems (NeurIPS 2022).

once every $1/\varepsilon$ steps, and (ii) the algorithm needs to *disambiguate information (correct predictions) from noise (incorrect predictions)*. We emphasize that we are interested in the "high noise" regime, i.e., small values of $\varepsilon$, where for every correct prediction, there are many incorrect ones. We will show that even in this regime, we can obtain online algorithms that are substantially better than the worst case. We also do not assume that the value of $\varepsilon$ is known to the online algorithm; rather, our algorithms automatically adapt to the (unknown) value of $\varepsilon$ in the underlying prediction model. In some cases, however, it will be easier to *describe* the algorithm and analysis if we assume that $\varepsilon$ is known. If that is the case, we will first present the simpler algorithm that knows $\varepsilon$ and then generalize it to the one that does not know $\varepsilon$.

**Prediction error vs $\varepsilon$-accurate predictions.** The trend in the online algorithms with predictions literature has been to use prediction error rather than prediction confidence to parameterize the performance of online algorithms. Note the difference in the two measures of prediction quality: an ML model with high confidence (i.e., that produces $\varepsilon$-accurate predictions for $\varepsilon$ close to 1) can occasionally make highly inaccurate predictions and therefore have very high prediction error; conversely, an ML model that has a small but consistent bias can have $\varepsilon = 0$ but still have small prediction error. So, the results obtained in this paper are complementary (and incomparable) to the existing literature on online algorithms parameterized by prediction error. To the best of our knowledge, neither caching nor covering problems have been studied in the setting of $\varepsilon$-accurate predictions.

Prior to our work, variants of $\varepsilon$-accurate predictions that also model confidence estimates of predictions have been studied in a few different problem settings such as frequency estimation in data streams [13], $k$-means [11], secretary problems [10], and page migration [14]. In some cases such as for ski rental [23] and set cover [6], the confidence parameter $\lambda$ does not represent a measure of correctness of the prediction, rather it is simply a hyperparameter that can be used to trade off the consistency and robustness bounds of the algorithm. In contrast to these, the goal in this paper is to understand the role of $\varepsilon$-accurate predictions in two central problem domains in online algorithms – caching and online covering problems.

**Competitive ratio.** As in previous literature in this area, we also use the classical performance metric of *competitive ratio* for online algorithms. The competitive ratio of an online algorithm is the worst-case ratio between the (expected) value of the objective in the algorithm's solution to that of the (offline) optimum across all input instances. In the $\varepsilon$-accurate predictions model, this expectation is taken over both the randomness of the algorithm (if any) and the randomness of the predictions. Our competitive ratios are expressed as a function of $\varepsilon$, and gracefully degrade as $\varepsilon$ goes from 1 to 0. But, even if $\varepsilon = 0$, we do not lose robustness; using standard techniques (e.g., [4]), we can combine the best online algorithm without predictions and our algorithm with $\varepsilon$-accurate prediction to match (up to constants) the better of the two solutions.

## 1.1 Our contributions

First, we consider the problem of *caching* with $\varepsilon$-accurate predictions. In this problem, there is an underlying (slow) main memory of $n$ pages; at any given point of time, some $k \ll n$ of these can be stored in a (fast) cache. The online input comprises a sequence of page requests $p_1, p_2, \ldots, p_T$, with page $p_t \in [n]$ being requested at time $t$. If page $p_t$ is not in the cache at this time, the algorithm must perform a *page swap*: it must *evict* some page from the cache, and *load* page $p_t$. The goal of the algorithm is the minimize the number of page swaps over the entire sequence of requests.

Lykouris and Vassilvitskii [18] initiated the study of caching with ML predictions and gave deterministic and randomized algorithms whose competitive ratio gracefully degrades with prediction error. (See [7, 15, 24, 27] for subsequent work on this problem.) The general idea in these algorithms is to evict the page whose next request is predicted to be *furthest in the future*, but robustify this strategy by appropriately combining it with a worst-case *randomized marking* algorithm if the predictions turn out to be inaccurate.

In our paper, we consider the caching problem with $\varepsilon$-accurate predictions. At each time, the algorithm is given a prediction for the FIF page in the cache that is correct with probability at least $\varepsilon$; with the remaining probability, the prediction is just a random page in the cache and hence, does not reveal any information about the future. A natural algorithm is to simply follow the prediction,

namely evict the predicted FIF page if the currently requested page is outside the cache. We call this the ONE-STRIKE algorithm and show that it achieves a competitive ratio of $O(1/\varepsilon)$. [1]

Our main result is to obtain an exponential improvement over this baseline. To achieve the better bound, we give a more nuanced algorithm that we call the TWO-STRIKES algorithm, and show that it achieves a competitive ratio of $O(\log(1/\varepsilon))$. We show this bound is tight by giving a matching lower bound of $\Omega(\log(1/\varepsilon))$ on the competitive ratio of any algorithm for caching with $\varepsilon$-accurate predictions when $\varepsilon = 1/k \log k$. (Note that as long as $\varepsilon > 0$, $\varepsilon$-accurate predictions are more accurate than predictions generated uniformly at random.)

Next, we consider the problem of solving online covering problems given $\varepsilon$-accurate predictions. Online covering is a general framework for many optimization problems such as set cover and network design. In this problem, the online algorithm has to maintain a solution on a fixed set of variables, and is only allowed to monotonically increase the variables over time. In each online step, the algorithm is presented with a new covering constraint on these variables, and must then augment its existing solution to satisfy the new constraint. The goal is to minimize a linear cost function defined on the variables.

The systematic study of online covering with predictions was initiated by Bamas, Maggiori, and Svensson [6], who used primal dual techniques to leverage a predicted optimal solution given to the online algorithm (see also [1] for subsequent work on online covering with predictions). We consider a model where in each online step, the algorithm gets a covering constraint and a predicted optimal way of satisfying the constraint. (Given an offline predicted solution, these online predictions can be generated on the fly.) However, these are only $\varepsilon$-accurate predictions. Namely, in each online step, the predicted solution to the covering constraint is part of an optimal solution with probability at least $\varepsilon$, and otherwise it is an *arbitrary* way of satisfying the constraint with the remaining probability. Note that in the latter case, the prediction does not reveal any information about the future. We give a simple algorithm for online covering with $\varepsilon$-accurate predictions that obtains a competitive ratio of $O(1/\varepsilon)$, and also a matching lower bound of $\Omega(1/\varepsilon)$. Our covering framework and algorithm are very flexible: they can even model and solve problems like facility location that are not covering problems in a strict sense.

**Related work.** There has been much recent work on incorporating ML predictions in online algorithms, such as in ad-allocation [19], auction pricing [20], page migration [14], flow allocation [17], scheduling [16, 22, 23], frequency estimation [13], speed scaling [5], Bloom filters [21], bipartite and secretary problems [3], and online linear optimization [9]. In particular, the caching problem was studied by [7, 15, 18, 24, 27], where the prediction model is that at each time $t$, the algorithm is given a prediction of when currently requested page $p_t$ is *next* requested, and the prediction error $\eta$ is defined as the $\ell_1$ error between the predicted and actual request times. Antoniadis *et al.* [2] consider the more general problem of metrical task systems; their predictor gives a state $p_t$ of the optimal solution, and the error $\eta$ is the sum of distances between the predictions and actual states. The online set cover problem with predictions was studied by [5], where the (offline) prediction provides an entire feasible solution at the outset. Their algorithm uses the online primal-dual framework, and uses a hyperparameter $\lambda \in [0, 1]$ to obtain a trade off between the consistency and robustness bounds.

## 1.2 Preliminaries

For every positive integer $n$, let $[n] := \{1, 2, \ldots, n\}$. Let us formalize the prediction model: at each time $t$, the algorithm is given some *prediction* or *suggestion* $s_t$, whose nature depends on the problem statement. At each time $t$, the algorithm is given one of two possible suggestions: a "good" suggestion $y_t$, or a "bad" suggestion $z_t$. While the precise definitions of $y_t$ and $z_t$ depend on the specific problems, we require that the former reveals some information about the optimal solution, and make no such requirement about the latter. There is a fixed value $\varepsilon \in (0, 1]$ that is unknown to the algorithm. Now at each time step $t$, the $\varepsilon$-*accurate prediction* $s_t$ is determined as follows, independently of the past:

$$s_t = \begin{cases} y_t & \text{with probability at least } \varepsilon \\ z_t & \text{otherwise.} \end{cases}$$

---

[1]More precisely, we can get $O(\min(1/\varepsilon, \log k)$ by combining ONE-STRIKE with Randomized Marking using standard techniques (see e.g., [2, 18]). As this applies to all our algorithms, we omit writing $O(\min(\cdot, \log k))$ for simplicity.

We aim to bound the competitive ratio of the algorithm in terms of the noise parameter $\varepsilon$. Note that the randomness that determines whether $s_t = y_t$ or $s_t = z_t$ is hidden from both the adversary, who generates the input sequence, and the algorithm.

## 2 Caching

In the *caching problem* there is an underlying (slow) main memory of $n$ pages; at any given point of time, some $k \ll n$ of these can be stored in the (fast) cache. The online input comprises a sequence of page requests $p_1, p_2, \ldots, p_T$, with page $p_t \in [n]$ being requested at time $t$. If page $p_t$ is already in the algorithm's cache at this time, the algorithm does not need to do anything, else it must perform a *page swap*: it must *evict* some page in the cache, and *load* the page $p_t$. The goal of the algorithm is the minimize the number of page swaps over the entire sequence of requests.

Given the request sequence up-front, a strategy that minimizes the number of page swaps is *Bélády's rule* [8]: at each time $t$, if the requested page $p_t$ is not in the cache, evict the page that is requested the *furthest in the future* (FIF) among the pages in the cache. An online algorithm does not know the future requests, so this is not implementable; instead policies such as LRU [26] and *Randomized Marking* [12] are used to circumvent this lack of foresight by using past requests to predict the future.

In contrast to the recent work which assumes a prediction for the next-arrival time of the page requested at time $t$ [7, 18, 25, 27], we consider a model where at every time $t$, the algorithm is provided with a prediction on the page in its cache that will be requested furthest in the future. The prediction $s_t$ is a *noisy prediction*: with probability $\varepsilon$ it is the FIF page, else it is a *random page* from the algorithm's cache. Using the notation from Section 1.2, $y_t$ is the FIF page in the algorithm's cache, and $z_t$ is a page chosen uniformly at random from the the algorithm's cache. This assumption of uniformity is only made for simplicity. We can in fact show (see Appendix A.2) that it suffices to assume that the probability that a page $p$ is incorrectly predicted as the FIF page is at most $1/k\varepsilon^c$ where $c$ is a constant.

Note the two extremes: if $\varepsilon = 1$ then we can follow Bélády's rule and be optimal; if $\varepsilon = 0$ then the predictions are just random pages in the cache (which can be generated without any knowledge about the future), and we get back the classic worst-case online setting.

Starting with the first request, we partition the input sequence into *phases*, where a phase is defined as a maximal contiguous subsequence of requests containing $k$ distinct pages. (The last phase might contain fewer than $k$ distinct pages because of the termination of the input sequence.) We note that our algorithms do not actually require suggestions to be precisely the FIF page; they work unchanged as long as the correct predictions provide a page that is not requested in the current phase. (The FIF page is just one such page.)

All our algorithms use the idea of a *cache reset* at the beginning of each phase (except the first): replace the contents of the cache with the $k$ pages requested in the previous phase. Since each page brought back in due to a cache reset must have been evicted after it was last requested in the just-ended phase, we get the following lemma.

**Lemma 2.1** (Cache Reset Overhead). *If an algorithm that performs cache resets at the beginning of each phase, the number of page swaps in any cache reset is at most the number of page swaps performed in the previous phase.*

These cache resets allow us to *localize* the description and analysis of our algorithms to a single phase, because the state of the cache at the beginning of the phase only depends on the input sequence, and is now independent of the algorithm. Furthermore, we partition the (at most) $k$ pages requested in each phase into two sets: *clean* pages are the ones that were not requested in the previous phase, and the rest are called *stale*. (All the $k$ requested pages in the first phase are considered clean.) Let $\Delta_i$ denote the number of clean pages in the $i^{th}$ phase.

### 2.1 ONESTRIKE: A Deterministic $O(1/\varepsilon)$-Competitive Algorithm

Our first algorithm ONESTRIKE using $\varepsilon$-accurate predictions is simple and deterministic, and obtains a competitive ratio of $O(1/\varepsilon)$. In the first phase, it fetches each requested page into the cache. After this point in time, the cache always remains full (with $k$ pages). Each subsequent phase starts with a

cache reset. Let $C_t$ denote the algorithm's cache contents at the end of timestep $t$. During the phase, the algorithm is the following:

(a) If the requested page $p_t$ is already in $C_{t-1}$, do nothing (i.e., it sets $C_t \leftarrow C_{t-1}$).

(b) Else if $p_t \notin C_{t-1}$, evict the predicted page $s_t$, i.e., set $C_t \leftarrow (C_{t-1} \setminus s_t) \cup \{p_t\}$.

**Theorem 2.2.** *The **ONESTRIKE** Algorithm is $O(1/\varepsilon)$-competitive.*

**Remark.** The **ONESTRIKE** algorithm and its analysis both apply assuming a weaker prediction model, in which the $z_t$ predictions (i.e., the non-good ones) are chosen adversarially, rather than uniformly at random among the pages in the algorithm's cache. Note that in this model, an adversary still does not know when good predictions are given, but can adapt to the algorithm's behavior when specifying the $z_t$ predictions.

## 2.2 TWOSTRIKES: A Randomized $O(\log 1/\varepsilon)$-Competitive Algorithm

We now use randomization to improve the competitive ratio to $O(\log(1/\varepsilon))$. Before describing the formal algorithm, we first give some intuition. The **ONESTRIKE** algorithm uses the prediction every time it must make a page swap. This places too much faith on the predictions, and incurs a large loss. Our first change is that the algorithm is more cautious with the predictions, and now views *two* predictions for the same page (and not just a single prediction) as a strong-enough signal to evict the page. We show that the probability that a page that remains the FIF page long enough is predicted twice far outweighs the probability that *any* non-FIF page is (erroneously) predicted twice during the course of the entire phase. As we now need two predictions before evicting a page, we need a fallback option if the requested page is not in the cache and no page has been predicted twice. We run a randomized marking algorithm (call it MARKER) for this purpose, and carefully combine these two algorithms to obtain the final algorithm.

However, this does not suffice: consider a situation where the length of the request sequence for which a page remains the FIF page in the cache is very short. (An extreme example is when pages are requested round-robin, in which case every page is the FIF page right after it is requested and remains so for only a single request.) In this case, none of the FIF pages maintain their FIF status long enough to be predicted twice, and the algorithm devolves to essentially being MARKER with a competitive ratio of $O(\log k)$. To handle this situation, we stop the algorithm once we are confident that we have seen at least $\Delta$ FIF pages in the predictions, where $\Delta$ is the number of clean pages requested in the phase. At this point, we switch to a different algorithm, which is again MARKER but run only on pages predicted earlier in the phase.

### 2.2.1 The TWOSTRIKES Algorithm

We now formally describe the **TWOSTRIKES** algorithm. (In this section we assume that we know the accuracy parameter $\varepsilon$: we remove this assumption later in the supplementary material.) As in **ONESTRIKE**, the first phase in **TWOSTRIKES** brings the first $k$ requested pages into the cache and does no evictions. Every subsequent phase begins with a *cache reset* to ensure that the pages in the cache are precisely those requested in the previous phase. We now describe the behavior of the **TWOSTRIKES** algorithm for a single phase.

**Epochs and Segments.** Since the algorithm does not know $\Delta$ (the number of clean requests in this phase), it maintains a *guess* $\widehat{\Delta}$, which starts at 1 and is periodically updated. These updates break the phase into *epochs*: the first epoch starts at the beginning of the phase; each time we observe that the number of clean requests in the phase exceeds our guess $\widehat{\Delta}$, we double the value of $\widehat{\Delta}$, thereby ending the current epoch and starting a new one.

When an epoch starts, **TWOSTRIKES** first performs a cache reset, and then checks if $\varepsilon \leq (\widehat{\Delta}/k)^{1/5}$; if so, it simply runs randomized marking in the rest of the current epoch. Else if $\varepsilon > (\widehat{\Delta}/k)^{1/5}$, the epoch now is partitioned into an *explore segment* followed by an *exploit segment*. In the explore segment, the algorithm makes $\widehat{\Delta}^*$ good evictions of pages that have been predicted twice (for some $\widehat{\Delta}^* \leq \widehat{\Delta}$), and also learns a small candidate set of pages that contains at least $\widehat{\Delta} - \widehat{\Delta}^*$ pages which make for good evictions. In the following exploit segment, the algorithm then runs randomized marking on these candidate pages to actually make $\widehat{\Delta} - \widehat{\Delta}^*$ good evictions.

Before we give details about these two segments, let us give two procedures: MARKER and STRIKER. The MARKER procedure maintains a binary flag called MARK for every page; we say that page $p$ is *marked* if MARK($p$) = 1, else it is *unmarked*. We mark and unmark pages by changing the flag to 1 or 0 respectively. All pages are unmarked at the beginning of the epoch. The MARKER algorithm essentially runs the RANDOM MARKING algorithm, but since we may run it starting with only a few unmarked pages, we allow for the possibility of running out of unmarked pages—in which case we declare failure. The details of MARKER appear in Algorithm 1.

---
**Algorithm 1:** MARKER

---
1.1   let $C$ be the set of pages in the cache
1.2   **if** *page $p_t$ is requested at the current time $t$* **do**
1.3      **case** $p_t \in C$ **do** do nothing
1.4      **case** $p_t \notin C$ *and* $|C| < k$ **do** $C \leftarrow C \cup \{p_t\}$
1.5      **case** $p_t \notin C$ *and* $|C| = k$ *and $C$ has at least one unmarked page* **do**
1.6         let $q_t$ be a uniformly random unmarked page in $C$, and set $C \leftarrow (C \setminus \{q_t\}) \cup \{p_t\}$
1.7      **otherwise do** declare FAIL
1.8   mark page $p_t$

---

The second procedure is STRIKER: it maintains a counter called STRIKE for every page, which takes on values in $\{0, 1, 2\}$. we say that page $p$ is *striked* if STRIKE($p$) $\in \{1, 2\}$, otherwise STRIKE($p$) = 0 and page $p$ is *unstriked*. If STRIKE($p$) = 2, we say page $p$ is *strike-evicted*. At the beginning of the epoch, all pages are unstriked. The STRIKER procedure operates in two modes: *active* and *passive*. The procedure is in the active mode when the cache contains $k$ pages (i.e., it is full), and it is in the passive mode (and does nothing) otherwise. Like MARKER, STRIKER is not a stand-alone caching algorithm; instead, it is active when the cache is full to acknowledge the prediction, and possibly performs a preemptive eviction. The details appear in Algorithm 2.

---
**Algorithm 2:** STRIKER

---
2.1   let $C$ be the set of pages in the cache
2.2   **if** $|C| < k$ **then** do nothing (we are in the passive mode)
2.3   **else if** *page $s_t \in C$ is the predicted page at the current time $t$* **then**
2.4      STRIKE($s_t$)++
2.5      **if** STRIKE($s_t$) = 2 **then** evict $s_t$, so $C \leftarrow C \setminus \{s_t\}$ (so that $s_t$ is *strike-evicted*)

---

**Switching to ONESTRIKE.** In both the explore and exploit segments, it is possible for the algorithm to switch to the **ONESTRIKE** algorithm. When this happens, all marks and strikes are forgotten, and the algorithm simply evicts the predicted page whenever an eviction is required. The only state that remains is the count on the number of clean pages in the phase so far: whenever it exceeds $\widehat{\Delta}$, regardless of the state of the algorithm, we double the value of $\widehat{\Delta}$ and start a new epoch.

**The Explore Segment.** We now describe our algorithm for the explore segment. It uses a global counter called BAD-STRIKES, initialized to 0 at the beginning of each epoch, that counts the number of bad evictions made by the STRIKER procedure. Recall that an eviction at time $t$ is good if the evicted page is not requested in the current phase after time $t$, and it is bad otherwise. Before serving each request, the algorithm first increments BAD-STRIKES as necessary, and then calls STRIKER and MARKER in that order. The formal description is in Algorithm 3. The explore segment ends if either (a) STRIKER makes at least $2\widehat{\Delta}$ evictions in the segment, or (b) STRIKER has been in active mode for $N := \widehat{\Delta}/\varepsilon^2$ requests.

**Lemma 2.3.** *In the explore segment, the* MARKER *procedure never declares* FAIL.

*Proof.* Pages are marked only when they are requested in the current epoch. So, if $p_t \notin C$ and $C$ has $k$ marked pages, then $k + 1$ distinct pages have been requested in the current epoch, and therefore also in the current phase. This contradicts the definition of a phase. $\square$

**The Exploit Segment.** The exploit segment ignores all predictions. Instead it relies on the fact (proved in Lemma 2.10) that the set of striked pages at the end of the explore segment contains $\widehat{\Delta}$

---
**Algorithm 3:** Explore Segment
---
3.1    **foreach** *time t* **do**

3.2      **let** $p_t$ be the requested page and $s_t$ the predicted page at time $t$

3.3      **if** $p_t$ *is strike-evicted (i.e.,* STRIKE$(p_t) = 2$) **then** increment BAD-STRIKES

3.4      **if** BAD-STRIKES $= \Delta$ **then** run **ONESTRIKE** for rest of epoch **else** set STRIKE$(p_t) \leftarrow 0$

3.5      **call** STRIKER

3.6      **call** MARKER

3.7      **terminate** explore segment if STRIKER evicts pages $\geq 2\widehat{\Delta}$ times, or STRIKER in active
       mode for $N := \widehat{\Delta}/\varepsilon^2$ requests

---

pages that would be good evictions (with good probability). The exploit segment now runs MARKER
on these striked pages. Formally, see Algorithm 4.

---
**Algorithm 4:** Exploit Segment
---
4.1    **mark** all pages in the cache with no strikes; striked pages are left unmarked

4.2    **foreach** *time t* **do**

4.3      **let** $p_t$ be page requested at time $t$

4.4      **if** $p_t$ *is strike-evicted (i.e.,* STRIKE$(p_t) = 2$) **then** increment BAD-STRIKES

4.5      **if** BAD-STRIKES $= \Delta$ **then** run **ONESTRIKE** for rest of epoch **else** set STRIKE$(p_t) \leftarrow 0$

4.6      **call** MARKER

4.7      **if** MARKER *returns* FAIL **then** run **ONESTRIKE** for the rest of the epoch

---

Since the exploit segment handles the possibility that MARKER may fail (and reverts to **ONESTRIKE**
in that case), the algorithm is well-defined; it only remains to bound the expected number of evictions
per epoch, and hence per phase. This is what we do next.

### 2.2.2   Competitive Ratio of the TWOSTRIKES Algorithm

**Theorem 2.4.** *The algorithm performs $O(\widehat{\Delta} \log 1/\varepsilon)$ evictions in expectation in an epoch.*

*Proof.* There are four types of evictions that happen during an epoch:

    (i) evictions performed by STRIKER in the explore segment,

   (ii) evictions performed by MARKER in the explore segment,

  (iii) evictions performed by MARKER in the exploit segment, and

  (iv) evictions due to the **ONESTRIKE** algorithm,

We show that the expected number of each of type of eviction is at most $O(\widehat{\Delta} \log 1/\varepsilon)$. The type (i)
evictions are the easiest: there are at most $2\widehat{\Delta}$ of these, by the termination condition of the explore
segment.

Next we bound the number of evictions of type (iii): consider running MARKER with some cache $C_0$,
where some $k - r$ pages are pre-marked. Consider the $r$ unmarked pages of $C_0$, and order them as
$p_1, p_2, \ldots, p_r$ in *reverse chronological order* of their first request in this segment. The pages that are
not requested at all are placed at the beginning of this sequence, but can be arbitrarily ordered relative
to each other. Let $D$ be the number of requests for pages outside the set $C_0$ received by the algorithm.
The following property holds by induction, since only $D$ pages are evicted, and each unmarked page
is equally likely to be evicted at any time:

**Lemma 2.5.** *At any time $t$, suppose pages $p_{i_t+1}, p_{i_t+2}, \ldots, p_r$ have already been requested and
pages $p_1, p_2, \ldots, p_{i_t}$ have not been requested yet. Then, for any $i \leq i_t$, the probability that page $p_i$
is not in the cache of the MARKER procedure at time $t$ is at most $\min(D/i_t, 1)$.*

Note that the property of this lemma is unconditional, in the sense that it does not depend on the set
of pages among $p_{i_t+1}, p_{i_t+2}, \ldots, p_r$ that were evicted before time $t$. The following corollary follows
by applying this lemma at the time of the first request for page $p_i$:

**Corollary 2.6.** *The probability that the cache of the* MARKER *procedure does not contain page $p_i$ at the time of its first request is at most* $\min(D/i, 1)$.

The next lemma is proved using the above corollary:

**Lemma 2.7.** *The expected number of evictions of type (iii) in an epoch is $O(\widehat{\Delta} \log(1/\varepsilon))$.*

*Proof.* The explore segment terminates after it sees at most $N = \widehat{\Delta}/\varepsilon^2$ requests for which STRIKER is in active mode. Let $S$ denote the set of striked pages; clearly, $|S| \leq N$. Moreover, the number of distinct clean page requests is at most $\widehat{\Delta}$, else the epoch ends. By Corollary 2.6, the expected number of evictions in $S$ is at most

$$\sum_{i=1}^{|S|} \min\left(\frac{\widehat{\Delta}}{i}, 1\right) \leq \widehat{\Delta} + \sum_{i=\widehat{\Delta}}^{N} \frac{\widehat{\Delta}}{i} = \widehat{\Delta} + \widehat{\Delta}(H_N - H_{\widehat{\Delta}}) = O(\widehat{\Delta} \log(1/\varepsilon)). \qquad \square$$

Bounding evictions of type (ii) is a bit more involved, because we start off with $k$ unmarked pages (so the naive application of Corollary 2.6 would give us $O(\widehat{\Delta} \log k)$); we need to use the interplay between MARKER and STRIKER to get our bound.

**Lemma 2.8.** *The expected number of evictions of type (ii) in an epoch is $O(\widehat{\Delta} \log(1/\varepsilon))$.*

Finally, to bound the number of type (iv) evictions, observe that we run **ONESTRIKE** if the counter BAD-STRIKES reaches $\widehat{\Delta}$, or if MARKER returns FAIL in the exploit segment. We claim that both of these events happen with probability at most $\varepsilon$. Now since **ONESTRIKE** makes $O(\widehat{\Delta}/\varepsilon)$ evictions in expectation the expected number of such evictions is at most $\varepsilon \cdot O(\widehat{\Delta}/\varepsilon) = O(\widehat{\Delta})$.

**Lemma 2.9.** $\Pr[\text{BAD-STRIKES} \geq \widehat{\Delta}] \leq \varepsilon$.

**Lemma 2.10.** *The probability that* MARKER *declares* FAIL *in the exploit segment is at most $O(\varepsilon)$.*

We now summarize the bounds on the four types of evictions. In the explore segment, STRIKER performs at most $2\widehat{\Delta}$ evictions by design of the algorithm, and MARKER performs $O(\widehat{\Delta} \log(1/\varepsilon))$ evictions by Lemma 2.8. In the exploit segment, MARKER performs $O(\widehat{\Delta} \log(1/\varepsilon))$ evictions by Lemma 2.7. Finally, Lemmas 2.9 and 2.10 show that **ONESTRIKE** is called with probability $O(\varepsilon)$, and its cost is $O(\widehat{\Delta}/\varepsilon)$, so its expected contribution is $O(\widehat{\Delta})$ evictions. Combining everything together, the total expected number of evictions in any epoch is at most $O(\widehat{\Delta} \log 1/\varepsilon)$, which proves Theorem 2.4. $\qquad \square$

Since we double our guess for $\widehat{\Delta}$ each time, the total expected cost of a phase is $O(\Delta \log 1/\varepsilon)$, thereby proving the claimed competitive ratio for the known-$\varepsilon$ case. The algorithm (and analysis) when we do not know the value of $\varepsilon$ in advance are conceptually similar to the one above, but there are more details to consider. In particular, we maintain a guess $\hat{\varepsilon}$ for $\varepsilon$, and each time we square this guess. The real problem is that unlike the value of $\Delta$, we do not get a clear signal that we have overestimated the value of $\varepsilon$. Our algorithm therefore needs to infer failure from our algorithm not performing as claimed; we defer this to the supplementary materials.

**Comparison with [27].** We now show that under our prediction model, the algorithm given by Wei [27] performs poorly. That algorithm combines RANDOM MARKING and BlindOracle (which in our terminology is called **ONESTRIKE**). Since the BlindOracle algorithm just evicts the page suggested by the oracle, we take $n = k + 1$ pages and construct a sequence of phases. In phase $i$ we request all pages except page $i$ round-robin, and do this $k$ times. The optimal strategy is to evict page $i$ at the start of this phase. But the algorithm follows the oracle blindly, so it will evict random pages due to bad suggestions $1/\varepsilon$ times in expectation before getting a good suggestion and evicting page $i$. This happens in each phase, giving an expected cost $\Omega(1/\varepsilon)$ times the optimum. Since this is combined with RANDOM MARKING which has an $\Omega(\log k)$ lower bound, by setting $\varepsilon = 1/\log k$ and interleaving phases of the above lower bound sequence with phases of the lower bound for RANDOM MARKING, we get a sequence that causes Wei's algorithm to pay $\Omega(\log k)$ times the optimal cost, whereas our algorithm pays $O(\log(1/\varepsilon)) = O(\log \log k)$ times the optimal cost.

# 3 Set Cover and Generalized Covering Problems

We now show that for a large class of online covering problems, which includes set cover (and its multiset multicover variants), facility location, and network design problems such as Steiner tree and Steiner forest, we can obtain an $O(1/\varepsilon)$-competitive algorithm with $\varepsilon$-accurate predictions. We first present it for the case of online set cover, and then extend the ideas to more general settings.

## 3.1 The Set Cover Problem

In the SET COVER problem we are given a collection of $m$ subsets $\mathcal{S} := \{S_1, \ldots, S_m\}$ of a universe $U$ containing $|U| = n$ elements. Each set has a cost $c(S) \geq 0$, and we want to pick a subcollection of least cost covering the universe. In the *online version* of the problem, we do not know the set system in advance: an element $e_t \in U$ is revealted at each time $t$, along with the names of the sets containing $e_t$. If none of these sets have already been chosen by the algorithm, one must be chosen at this time. Sets once chosen cannot be dropped (so the solutions are monotonically increasing), and the goal is to minimize the total cost of the chosen sets.

Our model of $\varepsilon$-accurate suggestions is the following: at each time, we are given an element $e_t$, along with a suggested set $s^t \in \mathcal{S}$, with the guarantee that $\Pr[s^t \in \mathsf{OPT} \mid \mathcal{H}^{t-1}] \geq \varepsilon$, where $\mathcal{H}^{t-1}$ is the history of all requests, predictions, and actions taken in previous steps, and $\mathsf{OPT}$ is the optimal offline solution.

> **SET-HEDGE**: If $e_t$ is already covered, do nothing. Else let $g^t$ be the minimum-cost set that covers $e_t$. Choose $s^t$ with probability $c(g^t)/c(s^t)$, and choose $g^t$ otherwise.

Since $g^t$ is the cheapest set covering $e_t$, we have $c(g^t)/c(s^t) \leq 1$ and hence a valid probability.

**Theorem 3.1.** *Given $\varepsilon$-accurate predictions, the* **SET-HEDGE** *algorithm is $2/\varepsilon$-competitive.*

This result is tight up to constant factors: we show in the supplementary material that no algorithm for set cover can have competitive ratio better than $O(1/\varepsilon)$ in general. Moreover, we can run this algorithm in parallel with any other online set cover algorithm, say one that is $\alpha_{SC}$-competitive, to get an algorithm that is $O(\min(1/\varepsilon, \alpha_{SC}))$-competitive.

## 3.2 Extension to Generalized Covering Problems

The simplicity of the algorithm allows us to extend to very general set of objective functions and constraints, which we call *generalized submodular-cost coverage* (GSCC). Consider the following:

1. The algorithm controls a point $x \in \mathbb{R}^d_{\geq 0}$. The initial point is $x^0 = \mathbf{0}$, the all-zeros vector. We require that $x$ is monotone over time; i.e., $x^{t-1} \leq x^t$.

2. (Covering.) At each time $t$, a set $K_t \subseteq \mathbb{R}^d_{\geq 0}$ is revealed, and we want that $x \in \cap_{s \leq t} K_s$. We restrict ourselves to sets that are closed under taking component-wise maximums—i.e., $x, y \in K_t \implies (x \vee y) \in K_t$, where $(x \vee y)_i = \max(x_i, y_i)$.

3. (Monotonicity and Submodularity.) The objective function $c : \mathbb{R}^d \to \mathbb{R}_{\geq 0}$ is *monotone*: if $x \leq y$ then $c(x) \leq c(y)$. Moreover, it is *submodular*: $c(x \vee y \vee z) - c(x \vee z) \leq c(x \vee y) - c(x)$.

The $\varepsilon$-accurate suggestion model now says: at each time $t$, the suggestion $s^t \in K_t$, and moreover $\Pr[s^t \leq x^* \mid \mathcal{H}^{t-1}] \geq \varepsilon$, where $x^*$ is the optimal offline solution. The **COVER-HEDGE** algorithm now extends the **SET-HEDGE** algorithm as follows:

> Let $g^t$ be the minimum-cost increment—i.e., $g^t \leftarrow \arg\min\{c(x^{t-1} \vee g) \mid g \in K_t\}$. Then set $x^t \leftarrow x^{t-1} \vee s^t$ with probability $\frac{c(x^{t-1} \vee g^t) - c(x^{t-1})}{c(x^{t-1} \vee s^t) - c(x^{t-1})}$, and $x^t \leftarrow x^{t-1} \vee g^t$ otherwise.

In general, finding this miminum-cost increment may be computationally hard; we focus on the information-theoretic considerations for now, and defer the computational issues for later.

**Theorem 3.2.** *Given $\varepsilon$-accurate suggestions, the* **COVER-HEDGE** *algorithm is $2/\varepsilon$-competitive for any generalized covering problem.*

### 3.3 Applications

Beyond set cover, the general covering formulation above captures several interesting problems:

**Network Design:** In the *Survivable Network Design* problem (which generalizes *Steiner Tree* and *Steiner Forest*) we are given a set $V$ of $n$ points together with the distances $d_{ij}$ between them $i, j \in V$. The goal is to connect $k$ *pairs* of points $\{(s_\ell, t_\ell) \in V \times V \mid \ell \in [k]\}$ at minimum cost, where each pair $(s_\ell, t_\ell)$ comes with a connectivity requirement of $r_\ell$ disjoint paths between them. This problem can be written as

$$\min\left\{ \sum_{i,j \in V} d_{ij} x_{ij} \mid \sum_{i \in S, j \notin S} x_{ij} \geq r_\ell \; \forall S, \ell \in [k] : s_\ell \in S, t_\ell \notin S; \; x \in \mathbb{R}_{\geq 0}^{n^2} \right\}.$$

This formulation satisfy the Covering, Monotonicity, and Submodularity properties. Moreover, the least-cost augmentation $g^t$ can be obtained in polynomial time by a minimum cost flow algorithm.

**Facility Location:** This is not a covering program, yet it can be modeled using our framework. Given a point set $V$ on $n$ points and distances $d_{ij}$ between points $i, j \in V$, and *opening costs* $f_i \geq 0$ for each $i \in V$, the goal is to designate some subset $F$ of points as facilities, so that the total cost of $\sum_{i \in F} f_i + \sum_{j \in V} \min_{i \in F} d_{ij}$ is minimized. We can write this using Balinski's MILP formulation:

$$\min\left\{ \sum_{i \in V} f_i y_i + \sum_{j \in V} d_{ij} x_{ij} \mid \sum_i x_{ij} = 1 \; \forall j \in V, \; x_{ij} \leq y_i \; \forall i, j \in V, y \in \mathbb{Z}_{\geq 0}^n, x \in \mathbb{R}_{\geq 0}^{n^2} \right\}.$$

It can be verified that this formulation also satisfies the Covering, Monotonicity, and Submodularity properties, despite the non-covering constraints of the form $x_{ij} \leq y_i$. (Indeed, in the supplementary material, we show that the coordinate-wise maximum covering property allows us to go beyond standard covering programs.) Moreover, the least-cost augmentation $g^t$ can be obtained in polynomial time using a simple greedy algorithm.

**Covering Mixed-Integer Linear Programs (MILPs):** given non-negative $A \in \mathbb{R}^{m \times n}$, $b \in \mathbb{R}^m$ and $c, u \in \mathbb{R}^n$, and a subset $I \subseteq [n]$, we want to solve

$$\min\{c^\intercal x \mid Ax \geq b, 0 \leq x \leq u, x_i \in \mathbb{Z} \, \forall i \in I, x_i \in \mathbb{R} \, \forall i \notin I\}.$$

Each constraint is a half-space, and it can be verified that this formulation satisfies the Covering, Monotonicity, and Submodularity properties. However, since this problem is NP-hard in general, the least-cost augmentation $g^t$ can be computed efficiently only in some cases such as the network design problems described above.

## 4    Conclusions

In this paper, we presented online algorithms augmented with $\varepsilon$-accurate predictions for several classic problems such as caching, set cover, facility location, Steiner tree, and generalizations. Can we show a a $\mathrm{poly} \log(1/\varepsilon)$ competitiveness, or even $f(\varepsilon)$ competitiveness for the $k$-server problem, which is an extension of our results for caching? Another direction would be to combine the $\varepsilon$-accuracy prediction model studied in this paper with some measure $\eta$ of total prediction error namely, design online algorithms augmented by *probabilistically approximately correct* (or PAC) predictions. Finally, a third direction of future work would be a thorough experimental analysis of our framework and algorithms, to empirically compare the performance of our work with that of other learning-augmented algorithms.

## Acknowledgments

AG and BS were supported in part by NSF awards CCF-1955785, CCF-2006953, and CCF-2224718. DP and KS were supported in part by NSF grants CCF-1750140 (CAREER) and CCF-1955703 and ARO grant W911NF2110230.

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
