**Organization.** Appendix A and Appendix B contain the missing proofs for Section 2, namely:

- Appendix A.1 contains the proofs for the **ONESTRIKE** algorithm from Section 2.1.
- Appendix A.2 contains the proofs required for Section 2.2.
- Appendix B describes and analyzes the **TWOSTRIKESUA** algorithm, which is an extension of the **TWOSTRIKES** to the case where the $\varepsilon$ is not known.
- Appendix B.2 proves the $\Omega(\log(1/\varepsilon))$ randomized lower bound.

Appendix C presents the missing proofs for Section 3, namely:

- Appendix C.1 contains the proofs of Theorem 3.1 and Theorem 3.2.
- Appendix C.2 contains the proof of Theorem 3.2 for the case where the objective function is separable.
- Appendix C.3 shows a syntactic characterization of generalized covering problems over half-spaces, which allows the extension of our results to problems such as facility location, as mentioned in Section 3.3.
- Appendix C.4 proves the $\Omega(1/\varepsilon)$ randomized lower bound.

# A    Missing Proofs from Section 2

First, we prove Lemma 2.1 (Cache Reset Overhead):

*Proof of Lemma 2.1.* Every page that is being brought into the cache during the cache reset must have been swapped out in the previous phase after serving its request. The lemma follows.    □

## A.1    Proofs for the ONESTRIKE algorithm

Next, we prove Theorem 2.2, the competitive ratio of the **ONESTRIKE** algorithm.

First, we introduce some helper lemmas. The first lemma is a standard property of caching that helps lower bound the cost of any feasible solution:

**Lemma A.1** (Lower Bound on the Optimal Cost). *The number of page swaps in any feasible solution (and therefore, in an optimal solution) is at least $\sum_{i \geq 2} \Delta_i/2$, where $\Delta_i$ is the number of clean pages requested in phase $i$.*

*Proof.* For any $i \geq 2$, a total of $k + \Delta_i$ distinct pages are requested across phases $i - 1$ and $i$, which means that any feasible solution must perform at least $\Delta_i$ page swaps across these two phases. The lemma follows by summing over all the values of $i$.    □

At any time $t$, recall that a *good eviction* is one where the evicted page is not requested in the current phase after time $t$. Once evicted, such a page does not return to the cache for the rest of the phase. This allows us to relate good evictions to clean requests:

**Lemma A.2** (Good Evictions vs. Clean Requests). *The number of good evictions in phase $i$ is at most $\Delta_i$, the number of clean pages requested in the phase.*

*Proof.* Note that a total of $k + \Delta_i$ pages are requested across phases $i - 1$ and $i$. After $\Delta_i$ good evictions in phase $i$, all subsequent requests in phase $i$ are for the remaining $k$ pages. The cache can hold all these $k$ pages, so there cannot be any further page swaps in phase $i$.    □

Finally, we relate good evictions to FIF pages:

**Lemma A.3** (Good Evictions vs. FIF Pages). *At any time $t$, if the cache contains $k$ pages and the FIF page is evicted to serve a request for a page not currently in the cache, then this is necessarily a good eviction.*

*Proof.* Excluding the currently requested page, at most $k-1$ other pages can be requested after time $t$ in the current phase. Now, note that the $k-1$ pages in the cache other than the FIF page are requested before the FIF page after time $t$. The lemma follows. $\qquad\square$

Now, we are ready to prove Theorem 2.2:

*Proof of Theorem 2.2.* By Lemma 2.1 and Lemma A.1, it suffices to prove that the expected number of page evictions of the algorithm in a given phase is at most $\Delta/\varepsilon$, where $\Delta$ is the number of clean pages requested in the phase. By Lemma A.3 and the $\varepsilon$-accuracy condition, every eviction is a good eviction with probability at least $\varepsilon$. It follows by Lemma A.2 that the expected number of page evictions made by the **ONESTRIKE** algorithm in the current phase is at most $\Delta/\varepsilon$. $\qquad\square$

## A.2 Proofs for the TWOSTRIKES algorithm

Now we provide the missing proofs for establishing the competitive ratio of the **TWOSTRIKES** algorithm.

*Proof of Lemma 2.8.* Suppose $p_1, p_2, \ldots, p_k$ are the unmarked pages in the cache at the beginning of the epoch in reverse chronological order of their first requests in this segment. The pages that are not requested at all are placed at the beginning of this sequence, but can be arbitrarily ordered relative to each other. As in Lemma 2.7, the number of clean requests for pages is at most $\widehat{\Delta}$, else the epoch ends. So by Corollary 2.6, the expected number of evictions of type (ii) among pages $p_1, p_2, \ldots, p_{2N}$ is again at most $O(\widehat{\Delta} \log(1/\varepsilon))$, using the same argument as in Lemma 2.7.

But what about evictions of type (ii) among the remaining pages $p_{2N+1}, \ldots, p_k$? Suppose at time $t$, the first request for page $p_{i_t}$ causes a type (ii) eviction of a page in $p_{2N+1}, \ldots, p_k$. At this point in time, the cache must be full (else we would not have evicted a page), so the STRIKER algorithm is active at the end of time $t$. By Lemma 2.5, the probability that some page $p_j$ has been evicted by MARKER by time $t$ is at most $\widehat{\Delta}/i_t \leq \widehat{\Delta}/2N = \varepsilon^2/2$ for any $j \in \{2N+1, \ldots, k\}$. By the union bound, none of the pages $p_j$ for $j \in \{2N+1, \ldots, k\}$ satisfying $j \in [i_t - 1/\varepsilon^2, i_t)$ have been evicted by MARKER by time $t$, with probability at least $1/2$.

Supposse this event happens. Since STRIKER is now active, one of the following events must happen in the next $1/\varepsilon^2$ requests:

$\mathcal{E}_a$: There is a type (i) eviction in these $1/\varepsilon^2$ requests, or
$\mathcal{E}_b$: The STRIKER procedure is active for all these $1/\varepsilon^2$ requests.

In either case, we can charge to the corresponding event. By the first termination condition for the explore segment, event $\mathcal{E}_a$ happens at most $2\widehat{\Delta}$ times. By second termination for the explore segment, event $\mathcal{E}_b$ happens at most $\widehat{\Delta}$ times. Putting these together, we conclude that the expected number of type (ii) evictions for pages $p_{2N+1}, \ldots, p_k$ is $O(\widehat{\Delta})$, and the total expected type (ii) evictions are at most $O(\widehat{\Delta} \log(1/\varepsilon))$. $\qquad\square$

*Proof of Lemma 2.9.* We show that $\mathbb{E}[\text{BAD-STRIKES}] \leq \varepsilon\widehat{\Delta}$, whereupon Markov's inequality implies the claimed probability bound. Indeed, fix any page $p$, and consider the expected number of times that the counter BAD-STRIKES is incremented due to $p$. For any such increment, $p$ must be predicted twice when it is not the FIF page in a cache containing $k$ pages. This is because if $p$ is the FIF page in any of these predictions, then by Lemma A.3, $p$ is a good eviction. By the termination condition for STRIKER, it is active mode for at most $N = \widehat{\Delta}/\varepsilon^2$ steps. So the expected contribution of $p$ to BAD-STRIKES is at most

$$\sum_{i \geq 1} i \cdot \binom{N}{2i} \cdot \frac{1}{k^{2i}} \leq \sum_{i \geq 1} \frac{1}{2^i} \cdot \left(\frac{N}{k}\right)^{2i} \leq \left(\frac{N}{k}\right)^2 < \left(\frac{\widehat{\Delta}}{\varepsilon^2 k}\right)^2 \quad \text{since } N = \frac{\widehat{\Delta}}{\varepsilon^2}.$$

By linearity of expectation, the expected value of the counter BAD-STRIKES is at most

$$\left(\frac{\widehat{\Delta}}{\varepsilon^2 k}\right)^2 \cdot k = \widehat{\Delta} \cdot \frac{1}{\varepsilon^4} \cdot \frac{\widehat{\Delta}}{k} < \varepsilon\widehat{\Delta}, \quad \text{since } \frac{\widehat{\Delta}}{k} < \varepsilon^5. \qquad\square$$

**Relaxing the prediction model.** The proof of Lemma 2.9 assumes that the probability that a page $p$ is predicted when it is not the FIF page in a cache containing $k$ pages is at most $1/k$. Now we show that the lemma still holds assuming this probability is at most $1/k\varepsilon^c$ for any $c > 0$. The proof is essentially the same, but we need to change a few constants. Now, the expected contribution of $p$ to BAD-STRIKES is at most

$$\sum_{i \geq 1} i \cdot \binom{N}{2i} \cdot \frac{1}{(k\varepsilon^c)^{2i}} \leq \sum_{i \geq 1} \frac{1}{2^i} \cdot \left(\frac{N}{k\varepsilon^c}\right)^{2i} \leq \left(\frac{N}{k\varepsilon^c}\right)^2.$$

By linearity of expectation, it suffices to show that this quantity is at most $\varepsilon\widehat{\Delta}/k$. This holds if we set $N = \widehat{\Delta}/\varepsilon^d$ and assume $\widehat{\Delta} < \varepsilon^{2c+2d+1}k$. (In the proof of Lemma 2.9, we assumed $c = 0$ and set $d = 2$.) Under these modifications, all of the other proofs still hold; this lemma was the only one relying on an upper bound on the probability of a page being incorrectly predicted as FIF.

*Proof of Lemma 2.10.* We call a strike *good* if it applies to a page that will not be requested in the rest of the phase. Let $S$ denote the number of pages struck by at least one good strike during the explore segment. It suffices to show $S \geq \widehat{\Delta}$ with probability $1 - O(\varepsilon)$. If this happens, then at the beginning of the exploit phase, there are at least $\widehat{\Delta}$ good pages that have either been struck or evicted, hence we would not run out of unmarked pages before the epoch ends.

Suppose the explore segment ends due to the first termination condition: STRIKER has made $2\widehat{\Delta}$ evictions. By Lemma 2.9, with probability $1 - \varepsilon$, at most $\widehat{\Delta}$ of these evictions are bad, so at least $\widehat{\Delta}$ of them are good, as desired.

Now suppose the explore segment ends due to the second termination condition: STRIKER has been in active mode for $N = \widehat{\Delta}/\varepsilon^2$ requests. Whenever STRIKER is in active mode, it strikes the predicted page (and possibly evicts it). Conditioned on the past, each strike is good with probability at least $\varepsilon$, so the expected number of good strikes is at least $\varepsilon N = \widehat{\Delta}/\varepsilon$. Some of these good strikes apply to the same page, but STRIKER evicts any page with two strikes, so among the good strikes, at most $\widehat{\Delta}$ of them apply to pages that already have a good strike. Thus, we have $\mathbb{E}[S] \geq \widehat{\Delta}/\varepsilon - \widehat{\Delta}$.

We now finish with a concentration bound; we want to show that $S < \widehat{\Delta}$ with probability $O(\varepsilon)$. Letting $\delta = 1 - 1/(1/\varepsilon - 1)$ and $\mu = \widehat{\Delta}/\varepsilon - \widehat{\Delta}$, by a standard Chernoff bound, we have

$$\Pr(S < \widehat{\Delta}) = \Pr(S < (1 - \delta)\mu) \leq \exp(-\delta^2\mu/2) \leq \exp((4 - 1/\varepsilon)/2) = O(\varepsilon),$$

assuming $\varepsilon$ is sufficiently small. $\qquad\square$

# B  Caching with $\varepsilon$-Accurate Predictions for Unknown $\varepsilon$

In this section, we formally describe the **TWOSTRIKESUA** algorithm for unknown accuracy $\varepsilon$. At a high level, the algorithm is similar to the one described for known $\varepsilon$: each phase begins with a cache reset to ensure that the pages in the cache are precisely those requested in the previous phase. Within a phase, there is an outer loop that iterates over *epochs*, and an inner loop that iterates over *blocks*. Each epoch has a fixed value of $\widehat{\Delta}$, which is initialized to 1 at the beginning of the phase. When the number of clean pages requested in an epoch exceeds $\widehat{\Delta}$, we end the epoch, perform a cache reset, and start a new epoch after doubling the value of $\widehat{\Delta}$. Each block fixes the value of $\hat{\varepsilon}$, which is initialized to $1/2$ (or any constant strictly less than 1) at the beginning of an epoch. The condition for ending a block and starting a new one is more complicated than that for an epoch because there is no direct way for the algorithm to detect if $\varepsilon < \hat{\varepsilon}$. We will describe this condition later as part of the internals of a block.

Within a block starts, **TWOSTRIKESUA** first checks if $\hat{\varepsilon} \leq (\widehat{\Delta}/k)^{1/5}$; if so, it simply runs randomized marking in the rest of the current epoch. Else if $\hat{\varepsilon} > (\widehat{\Delta}/k)^{1/5}$, the block now is partitioned into an *explore segment* followed by an *exploit segment*. In the explore segment, the algorithm makes $\widehat{\Delta}^*$ good evictions of pages that have been predicted twice (for some $\widehat{\Delta}^* \leq \widehat{\Delta}$), and also learns an additional candidate set of pages that contains at least $\widehat{\Delta} - \widehat{\Delta}^*$ pages which would be good evictions. In the following exploit segment, the algorithm runs randomized marking on these candidate pages to actually make $\widehat{\Delta} - \widehat{\Delta}^*$ good evictions.

The algorithm requires the procedures MARKER and STRIKER. Both are identical to their respective counterparts from Section 2, where $\varepsilon$ was known. The explore and exploit segments are largely identical to their counterparts as well, with minor changes that we now describe.

**The Explore Segment.** The explore segment is almost identical to its counterpart from Section 2. The only difference is the following: If BAD-STRIKES (initially 0 at the beginning of each block) reaches $\widehat{\Delta}$, then we end the current block, perform a cache reset, and start a new block after squaring the value of $\hat{\varepsilon}$. The details are in Algorithm 5.

---

**Algorithm 5:** Explore Segment

---

**5.1** **foreach** *time $t$* **do**
**5.2**     **let** $p_t$ be the requested page and $s_t$ the predicted page at time $t$
**5.3**     **if** $p_t$ *is strike-evicted (i.e.,* STRIKE$(p_t) = 2$*)* **then** increment BAD-STRIKES
**5.4**     **if** BAD-STRIKES $= \widehat{\Delta}$ **then** square $\hat{\varepsilon}$ and start a new block **else** set STRIKE$(p_t) \leftarrow 0$
**5.5**     **call** STRIKER
**5.6**     **call** MARKER
**5.7**     **terminate** explore segment if STRIKER evicts pages $\geq 2\widehat{\Delta}$ times, or STRIKER in active
          mode for $N := \widehat{\Delta}/\hat{\varepsilon}^2$ requests

---

**The Exploit Segment.** The exploit segment is also almost identical to its counterpart from Section 2. As was the case for the explore segment, the only difference is that if BAD-STRIKES reaches $\widehat{\Delta}$, then we end the current block, perform a cache reset, and start a new block after squaring the value of $\hat{\varepsilon}$. The details are in Algorithm 6.

---

**Algorithm 6:** Exploit Segment

---

**6.1** **let** $S \leftarrow$ striked pages from the explore segment
**6.2** **mark** all pages in the cache not in $S$; pages in $S$ are unmarked
**6.3** **if** $p_t$ *requested at time $t$* **then**
**6.4**     **if** $p_t$ *is strike-evicted (i.e.,* STRIKE$(p_t) = 2$*)* **then** increment BAD-STRIKES
**6.5**     **if** BAD-STRIKES $= \widehat{\Delta}$ **then** square $\hat{\varepsilon}$ and start a new block **else** set STRIKE$(p_t) \leftarrow 0$
**6.6**     **call** MARKER **if** MARKER *returns* FAIL **then** square $\hat{\varepsilon}$ and start a new block

---

Note that the **TWOSTRIKESUA** algorithm, unlike **TWOSTRIKES** from Section 2, does not call **ONESTRIKE** at any point. Instead, whenever BAD-STRIKES reaches $\widehat{\Delta}$, it squares $\hat{\varepsilon}$ and starts a new block.

### B.1 Competitive ratio of the TWOSTRIKESUA algorithm

In any block, there are three types of page evictions in the randomized algorithm:

   (i) evictions performed by STRIKER in the explore segment,
  (ii) evictions performed by MARKER in the explore segment, and
 (iii) evictions performed by MARKER in the exploit segment

We can bound all three types using the same proofs from from Section 2, except we replace $\varepsilon$ with $\hat{\varepsilon}$. Furthermore, the page swaps during the cache reset at the beginning of a block can be charged to the page evictions in the previous block. So, we have arrived at the following bound:

**Lemma B.1.** *The expected number of evictions in a block is $O(\widehat{\Delta} \cdot \log(1/\hat{\varepsilon}))$.*

Now we address the main difference between **TWOSTRIKESUA** and **TWOSTRIKES**, which is the block structure due to our guess $\hat{\varepsilon}$ of $\varepsilon$. At a high level, Lemma B.1 allows us to bound the cost incurred in blocks where $\hat{\varepsilon} > \varepsilon$, because squaring $\hat{\varepsilon}$ creates a geometric series over these blocks. For blocks where $\hat{\varepsilon} < \varepsilon$, we now show that the probability of starting a new block is at most $2\hat{\varepsilon}$. This will in fact allow us to bound the total cost over these blocks by $O(\widehat{\Delta})$.

**Lemma B.2.** $\Pr[\text{BAD-STRIKES} \geq \widehat{\Delta}] \leq \hat{\varepsilon}$.

*Proof.* The proof is identical to that of Lemma 2.9, except with $\hat{\varepsilon}$ instead of $\varepsilon$. $\square$

**Lemma B.3.** *In any block where $\hat{\varepsilon} \leq \varepsilon$, the probability that MARKER declares FAIL is $O(\hat{\varepsilon})$.*

*Proof.* The proof is identical to that of Lemma 2.10, except with $\hat{\varepsilon}$ instead of $\varepsilon$. More specifically, suppose the explore segment ends due to the first termination condition (i.e., STRIKER has made $2\widehat{\Delta}$ evictions). Then by Lemma B.2, with probability $1 - \hat{\varepsilon}$, at most $\widehat{\Delta}$ of these evictions are bad so at least $\widehat{\Delta}$ of them are good, as desired.

Now suppose the explore segment ends due to the second termination condition: STRIKER has been in active mode for $N = \widehat{\Delta}/\hat{\varepsilon}^2$ requests. Whenever STRIKER is in active mode, it strikes the predicted page (and possibly evicts it). Conditioned on the past, each strike is good with probability at least $\varepsilon$, so the expected number of good strikes is at least $\varepsilon N = \varepsilon\widehat{\Delta}/\hat{\varepsilon}^2 \geq \widehat{\Delta}/\hat{\varepsilon}$. The rest of the proof is identical to that of Lemma 2.10, except with $\hat{\varepsilon}$ instead of $\varepsilon$. $\square$

**Lemma B.4.** *If $\hat{\varepsilon} \leq \varepsilon$, then the probability that the algorithm starts a new block is at most $O(\hat{\varepsilon})$.*

*Proof.* There are two ways for the algorithm to start a new block: if BAD-STRIKES $= \widehat{\Delta}$, or MARKER returns FAIL due to a request to a page not in the cache when the cache has $k$ marked pages. By Lemma B.2 the former occurs with probability at most $\hat{\varepsilon}$, and by Lemma B.3, the latter occurs with probability $O(\hat{\varepsilon})$. The lemma follows by a union bound. $\square$

Finally, we are ready to prove the competitive ratio of the algorithm.

**Lemma B.5.** *The competitive ratio of* **TWOSTRIKESUA** *is $O(\log(1/\varepsilon))$.*

*Proof.* We need to show that the expected number of evictions in a phase is $O(\Delta \log(1/\varepsilon))$. Since $\widehat{\Delta} \leq 2\Delta$ in each epoch and doubles between consecutive epochs in the same phase, it suffices to show that the expected number of page swaps in an epoch is $O(\widehat{\Delta} \log(1/\varepsilon))$. By Lemma B.1, the expected number of evictions in all blocks of a single epoch satisfying $\hat{\varepsilon} \geq \varepsilon^2$ is bounded by $O(\widehat{\Delta} \log(1/\varepsilon))$. By Lemma B.4, for any block with $\hat{\varepsilon} < \varepsilon$, the probability that the algorithm starts a new block is $O(\hat{\varepsilon})$. Thus, the expected number of swaps due to all blocks satisfying $\hat{\varepsilon} < \varepsilon^2$ is at most

$$O(\widehat{\Delta}) \cdot \sum_{i=1}^{\infty} \varepsilon^{2^i} \cdot \log \frac{1}{\varepsilon^{2^{i-1}}} \leq O(\widehat{\Delta}) \cdot \sum_{i=1}^{\infty} (2\varepsilon^2)^i \log \frac{1}{\varepsilon} = O(\widehat{\Delta}). \qquad \square$$

## B.2 Caching Lower Bound

**Theorem B.6.** *Any (randomized) algorithm for caching with $\varepsilon$-accurate suggestions is $\Omega(\log(1/\varepsilon))$-competitive.*

In our lower bound construction, we will consider the following prediction model: at each time $t$, with probability $\varepsilon$ (independently of the past), the algorithm receives a good prediction (i.e., it is told the FIF page in its cache). With probability $1 - \varepsilon$, it does not receive any prediction at all. This prediction model is stronger than $\varepsilon$-accurate suggestions, because any algorithm, given the former, can generate the latter by choosing a page uniformly at random from its cache in the $1 - \varepsilon$ case. Thus, a lower bound in this model also holds for the $\varepsilon$-accurate prediction model.

*Proof.* Let $n = k + 1$ and consider the sequence that generates each request uniformly at random among the $k + 1$ pages. Set $\varepsilon = \frac{1}{k \ln k}$. A *phase* is defined as a maximal contiguous subsequence of requests that contains exactly $k$ distinct pages. Consider an arbitrary phase and let $X$ be the random variable denoting its length. Note that $\mathbb{E}[X]$ is the expected number of times we would need to sample from a uniform random variable over a space of size $k + 1$ until we obtain $k$ distinct outcomes. By a slight modification of the coupon collector analysis, we can show that $X = \Theta(k \log k)$ with constant probability.

Now partition the input into groups of 3 consecutive phases (starting at the beginning), and consider any such group. Notice that every phase contains all but one of the $k + 1$ pages, and that missing page is the first page requested in the subsequent phase. So for each page $p$ suggested to the algorithm during the first phase, $p$ is either requested in the second phase, or $p$ is the first request of the third phase.

Furthermore, the probability that the algorithm does not receive any good suggestions during the second or third phase is $(1 - \varepsilon)^{\Theta(k \log k)} = \Theta(1)$. So overall in this group, with constant probability, the algorithm only receives good suggestions in the first phase, and these suggestions do not reveal

anything about the third phase (except for possibly the first request). So when serving the third phase of the group, the algorithm incurs a miss at every step with probability $1/(k+1)$, for an expected cost of $\Omega(\log k)$ to serve the group. On the other hand, the optimal solution can serve each phase by evicting the single page that does not appear in that phase, thereby incurring $O(1)$ cost per group. $\quad\square$

## C  Missing Proofs from Section 3

### C.1  Analysis of SET-HEDGE and COVER-HEDGE

*Proof of Theorem 3.1.* Fix any instance of online set cover and an optimal solution OPT for it. Let $\mathsf{ALG}^t$ be the solution of the **SET-HEDGE** algorithm after covering $e_t$. Consider the potential function

$$\Phi^t := (2/\varepsilon) \cdot \textstyle\sum_{S \in \mathsf{OPT} \setminus \mathsf{ALG}^t} c(S),$$

the cost of optimal sets not already picked by the algorithm by time $t$ (and scaled by $2/\varepsilon$). If $\Delta\Phi^t$ and $\Delta c(\mathsf{ALG}^t)$ denote the change in potential and the algorithm's cost due to sets picked at time $t$, we claim

$$\mathbb{E}[\Delta\Phi^t + \Delta c(\mathsf{ALG}^t)] \leq 0; \tag{1}$$

and because $\Phi^t$ is always non-negative, this shows that the total expected cost of sets picked by the algorithm is at most $\Phi^0 = (2/\varepsilon) \cdot c(\mathsf{OPT})$.

To prove (1), condition on $\mathcal{H}^{t-1}$: if $e_t$ is already covered by $\mathsf{ALG}^{t-1}$ we have $\Delta\Phi^t = \Delta c(\mathsf{ALG}^t) = 0$. Else if $e_t$ is not previously covered, $\Delta c(\mathsf{ALG}^t) = \frac{c(g^t)}{c(s^t)} \cdot c(s^t) + (1 - \frac{c(g^t)}{c(s^t)}) \cdot c(g^t) \leq 2\, c(g^t)$. The potential change is

$$\mathbb{E}[\Delta\Phi^t \mid \mathcal{H}^{t-1}] = -(2/\varepsilon) \cdot \sum_{S \in \mathsf{OPT} \setminus \mathsf{ALG}^{t-1} : e_t \in S} c(S) \cdot \Pr[s^t = S \mid \mathcal{H}^{t-1}] \cdot \frac{c(g^t)}{c(S)}.$$

By the $\varepsilon$-accuracy condition, the probability values sum to at least $\varepsilon$. Hence $\mathbb{E}[\Delta\Phi^t \mid \mathcal{H}^{t-1}] \leq -(2/\varepsilon) \cdot c(g^t) \cdot \varepsilon$. So (1) holds irrespective of $\mathcal{H}^{t-1}$ (and hence unconditionally). $\quad\square$

*Proof of Theorem 3.2.* The proof closely mimics that of Theorem 3.1; the potential is now

$$\Phi^t := (2/\varepsilon) \cdot \left[ c(x^* \vee x^t) - c(x^t) \right].$$

This is clearly non-negative (by monotonicity), and starts off at $\Phi^0 \leq (2/\varepsilon) \cdot c(x^*)$. Again it suffices to show that $\mathbb{E}[\Delta\Phi^t + \Delta c(\mathsf{ALG}^t)] \leq 0$. First, a direct calculation shows

$$\mathbb{E}[\Delta\, c(\mathsf{ALG}^t)] \leq 2 \cdot (c(x^{t-1} \vee g^t) - c(x^{t-1})).$$

To bound the expected potential decrease, for any vector $v$, setting $x^t \leftarrow x^{t-1} \vee v$ decreases the potential by

$$(2/\varepsilon) \cdot \left[ c(x^* \vee x^{t-1}) - c(x^{t-1}) - (c(x^* \vee x^{t-1} \vee v) - c(x^{t-1} \vee v)) \right],$$

which is non-negative by submodularity. Let $\mathcal{E}^t$ be the event that $p^t \leq x^*$, and let $\mathcal{F}^t$ be the event that we make the choice of setting $x^t \leftarrow x^{t-1} \vee p^t$. If both events happen, we have $c(x^* \vee (x^{t-1} \vee p^t)) = c(x^* \vee x^{t-1})$, and the expression above becomes

$$(2/\varepsilon) \cdot (c(x^{t-1} \vee p^t) - c(x^{t-1})).$$

Hence

$$\mathbb{E}[\Delta\Phi^t \mid \mathcal{H}^{t-1}] \leq -(2/\varepsilon) \cdot (c(x^{t-1} \vee p^t) - c(x^{t-1})) \cdot \Pr[\mathcal{E}^t \cap \mathcal{F}^t \mid \mathcal{H}^{t-1}].$$

Moreover, the two events are independent. By the $\varepsilon$-accuracy property, we know that $\Pr[\mathcal{E}^t \mid \mathcal{H}^{t-1}] \geq \varepsilon$. Finally, substituting the probability of $\mathcal{F}^t$ from the algorithm description, we get $\mathbb{E}[\Delta\Phi^t + \Delta c(\mathsf{ALG}^t) \mid \mathcal{H}^{t-1}] \leq 0$, which proves the claim. $\quad\square$

## C.2 Separable Objective Functions

We consider the same setting as Section 3.2, with the only difference being that instead of requiring that the objective function $c : \mathbb{R}^d \to \mathbb{R}_{\geq 0}$ is submodular, in this section, we require it to be separable. In particular, we assume there exist functions $c_1, \ldots, c_d : \mathbb{R} \to \mathbb{R}_{\geq 0}$ such that every $c_i$ is monotone, and for all $x \in \mathbb{R}^d$, $c(x) = \sum_i c_i(x_i)$. Note that this implies $c$ itself is monotone.

Recall the **COVER-HEDGE** algorithm, given a subset $K_t$ and suggestion $p^t$:

> Let $g^t$ be the minimum-cost increment—i.e., $g^t \leftarrow \arg\min\{c(x^{t-1} \vee g) \mid g \in K_t\}$.
> Then set $x^t \leftarrow x^{t-1} \vee p^t$ with probability $\frac{c(x^{t-1} \vee g^t) - c(x^{t-1})}{c(x^{t-1} \vee p^t) - c(x^{t-1})}$, and $x^t \leftarrow x^{t-1} \vee g^t$
> otherwise.

We claim that it is still $2/\varepsilon$-competitive in this setting. Note that when $c$ is separable, for any vectors $x, v \in \mathbb{R}^d$, we have $c(x \vee v) = \sum_i c_i(x_i \vee v_i)$.

**Theorem C.1.** *Given $\varepsilon$-accurate suggestions, the **COVER-HEDGE** algorithm is $2/\varepsilon$-competitive for any generalized covering problem where the objective is separable.*

*Proof.* Like that of Theorem 3.2, this proof closely mimics that of Theorem 3.1. The potential is the same as the one used to prove Theorem 3.2:

$$\Phi^t := \left(\frac{2}{\varepsilon}\right) \cdot \left[c(x^* \vee x^t) - c(x^t)\right]$$

$$= \left(\frac{2}{\varepsilon}\right) \cdot \left[\sum_{i=1}^{d} c_i(x_i^* \vee x_i^t) - c_i(x_i^t)\right]$$

$$= \left(\frac{2}{\varepsilon}\right) \cdot \left[\sum_{i : x_i^* \geq x_i^t} c_i(x_i^*) - c_i(x_i^t)\right].$$

This is non-negative (by monotonicity) and starts off at $\Phi^0 \leq (2/\varepsilon) \cdot c(x^*)$. Again it suffices to show that $\mathbb{E}[\Delta\Phi^t + \Delta c(\mathsf{ALG}^t)] \leq 0$. To bound the expected potential decrease, for any vector $v$, setting $x^t \leftarrow x^{t-1} \vee v$ decreases the potential by

$$(2/\varepsilon) \cdot \left[c(x^* \vee x^{t-1}) - c(x^{t-1}) - (c(x^* \vee x^{t-1} \vee v) - c(x^{t-1} \vee v))\right]$$

$$= (2/\varepsilon) \cdot \left[\sum_{i : x_i^* \geq x_i^{t-1}} c_i(x_i^*) - c_i(x_i^{t-1}) - \left(\sum_{i : x_i^* \geq x_i^{t-1} \vee v_i} c_i(x_i^*) - c_i(x_i^{t-1} \vee v_i)\right)\right],$$

which is non-negative since $x_i^* \geq x_i^{t-1} \vee v_i$ implies $x_i^* \geq x_i^{t-1}$, and every $c_i$ is monotone. The remainder of the proof is identical to that of Theorem 3.2. $\square$

## C.3 Properties of Generalized Covering Problems

The component-wise maximum property is satisfied for covering programs (i.e., where the constraints are $a^\mathsf{T} x \geq b$, and all entries of $a, b$ are non-negative). However, we now show that the property is also satisfied when $a$ has one negative coordinate.

**Fact C.2.** *Suppose $a \in \mathbb{R}^n, b \in \mathbb{R}$, and consider the set $K = \{x : a^\mathsf{T} x \geq b, x \geq 0\}$. Assume $K$ contains at least one non-zero vector. Then $K$ is closed under max if and only if $a$ has at most one negative coordinate.*

*Proof.* We first prove the backward direction. Let $x, y \in \mathbb{R}^n$ satisfy $a^\mathsf{T} x \geq b$ and $a^\mathsf{T} y \geq b$, let $z = x \vee y$. If $a$ has no negative coordinates, then clearly $a^\mathsf{T} z \geq b$. Now suppose there exists $i$ such that $a_i < 0$. Then we have

$$-a_i z_i + b \leq \max\left(\sum_{j \neq i} a_j x_j, \sum_{j \neq i} a_j y_j\right) \leq \sum_{j \neq i} a_j z_j,$$

which implies $a^\intercal z \geq b$, as desired.

For the forward direction, if $n = 1$, the statement is trivial. So for contradiction, without loss of generality, we assume $a_1, a_2 < 0$. Let $\alpha_i = -a_i$, $N = \{i : a_i < 0\}$, and $P = \{i : a_i > 0\}$. Now the condition $a^\intercal x \geq b$ is equivalent to

$$\alpha_1 x_1 + \alpha_2 x_2 + \sum_{i \in N \setminus \{1,2\}} \alpha_i x_i + b \leq \sum_{i \in P} a_i x_i. \tag{2}$$

Consider the following cases:

1. $P = \emptyset$: In this case, if $b \geq 0$, then the zero vector is the only one that could possibly satisfy (2), contradicting the assumption that $K$ contains at least one non-zero vector. If $b < 0$, then consider $u, v \in \mathbb{R}^n$ with all zeroes except $u_1 = -b/\alpha_1$ and $v_2 = -b/\alpha_2$. Then $w = u \vee v$ does not satisfy (2), since the left-hand side is $-b - b + b = -b > 0$ while the right-hand side is $0$.

2. $P \neq \emptyset$: Define $u, v \in \mathbb{R}^n$ such that the following conditions hold: $u_i = v_i = 0$ for all $i \in N \setminus \{1, 2\}$, $u_i = v_i$ for all $i \in P$, $(u_1, u_2) = (\varepsilon/\alpha_1, 0)$, $(v_1, v_2) = (0, \varepsilon/\alpha_2)$, and $\sum_{i \in P} a_i u_i = \sum_{i \in P} a_i v_i = b + \varepsilon$ for some arbitrary $\varepsilon > |b|$ (so $b + \varepsilon > 0$). Then $w = u \vee v$ does not satisfy (2), since the left-hand side is $\varepsilon + \varepsilon + b = b + 2\varepsilon$ while the right-hand side is $b + \varepsilon$ and $\varepsilon > 0$. $\qquad\square$

## C.4 Lower Bound for Set Covering

**Theorem C.3.** *Any algorithm for online set cover with $\varepsilon$-accurate suggestions is $\Omega(1/\varepsilon)$-competitive, even for the fractional case (or allowing randomization).*

As we did for the proof of Theorem B.6, we will consider the following prediction model: at each time $t$ (independently of the past), the algorithm receives a good prediction (i.e., a set included in the optimal solution) with probability $\varepsilon$. With probability $1 - \varepsilon$, it does not receive any prediction at all. Again, the algorithm can generate predictions on its own in the $1 - \varepsilon$ case.

*Proof.* Within $1/\varepsilon$ steps, there is a constant probability that the algorithm receives no suggestion. So we define a sequence of $1/\varepsilon$ requested elements as follows: the first element $e_1$ is contained in $2^{1/\varepsilon}$ sets. Each subsequent element $e_t$ is contained in exactly half of the sets that contain $e_{t-1}$, chosen uniformly at random. In expectation, the algorithm spends a total of $1/2$ on sets that contain $e_t$ and do not contain $e_{t'}$ for any $t' > t$. Since these spent amounts are disjoint, over $1/\varepsilon$ steps, the algorithm spends $\log(2^{1/\varepsilon})/2 = \varepsilon/2$ in expectation, while an optimal solution spends $1$ by picking the single set containing $1/\varepsilon$ elements.

We can scale this construction by making disjoint copies of this instance, and in each copy, there is a constant probability that the algorithm spends $\Omega(1/\varepsilon)$ while the optimal solution spends $1$. $\qquad\square$