# OpenReview forum: "Augmenting Online Algorithms with $\varepsilon$-Accurate Predictions"
_NeurIPS.cc/2022/Conference — NeurIPS 2022 Accept_

### Official Review · Reviewer_nuaT · 2022-07-01

**Rating:** 7
**Confidence:** 3
**Soundness:** 3 good
**Presentation:** 3 good
**Contribution:** 3 good

**Summary:**


The paper considers the problem of designing ML augmented online algorithms for caching and set cover problems. They initiate the study of these algorithms with eps-accurate predictions, that is algorithms which provide relevant information with probability eps, and otherwise provide arbitrary predictions.  More precisely, they focus on the “high noise” regime where eps is very small and predictions are for me for the most part uninformative.

Beginning with the online caching problem, the start by introduce an algorithm OneStrike which is O(1 / eps) competitive and later show how the algorithm can be refined into the TwoStrikes algorithm which is O(log(1/ eps) *  Delta) competitive where Delta is a problem parameter capturing the number of “clean” page requests. They then prove that no algorithm can do better than a Omega( log( 1/ eps)) ratio.

For the set-cover problem, they introduce algorithms with achieve a O(1/ eps) ratio which is again shown to be tight.


**Questions:**

I’ve included most of my questions above, my main concern is around this Delta parameter.
Very minor comment: add the FIF abbreviation on L80 where you first refer to the furthest in the future concept.


**Limitations:**

Limitations are appropriately discussed.

**Strengths And Weaknesses:**

The learning model considered in this paper is well-motivated and, to the best of my knowledge, novel. The ideas are for the most part very well presented. The idea of eps-accurate predictions is clearly introduced and clear. It would be valuable however to include more formal definitions in the main body of the paper. For example, the definition of a competitive ratio is defined in L62-64 in words, however, it would help with understanding if the authors provide a precise, formal definition of this quantity in the context of the caching and set cover problems they consider.

Apart from the competitive ratio, there are a number of other quantities which could benefit from having more formal definitions. For example, for caching: phases, and the number of clean pages should have more precise definitions.

From examining the upper and lower bounds, it looks like their results are unimprovable up to this factor of \Delta which captures the number of clean pages requested in a phase. It wasn’t clear to me how to think about this parameter and its relationship to the size of the cache k. Adding further discussion on this point would be helpful.

From a significance perspective, the paper appears to make a valuable contribution to the growing literature on learning augmented algorithms. The results are nearly tight for the problems they consider and provide a useful way of leveraging the eps-accurate prediction model which they introduce.

---

> ### Author Response · Authors · 2022-08-02
> **Response to nuaT**
>
> > "Apart from the competitive ratio, there are a number of other quantities which could benefit from having more formal definitions. For example, for caching: phases, and the number of clean pages should have more precise definitions."
>
> Thanks, we will add clear definitions for the various quantities.
>
> > "From examining the upper and lower bounds, it looks like their results are unimprovable up to this factor of $\Delta$ which captures the number of clean pages requested in a phase. It wasn’t clear to me how to think about this parameter and its relationship to the size of the cache k. Adding further discussion on this point would be helpful.
>
> Indeed, we don't want to depend on $\Delta$ (or $\varepsilon$), so the proofs in the supplementary material show how to get away from knowing either $\Delta$ or $\varepsilon$, by using a guess-and-double-based estimation procedure. We will clarify this further in the next version.

---

> > ### Comment · Reviewer_nuaT · 2022-08-07
> > **Thanks for the response.**
> >
> > I've read it and it helped clear up my questions.

---

### Official Review · Reviewer_iZrF · 2022-07-07

**Rating:** 6
**Confidence:** 4
**Soundness:** 3 good
**Presentation:** 3 good
**Contribution:** 2 fair

**Summary:**

This paper introduces a new model of learning-augmented online algorithms and studies caching and generalized covering problems in this model. In this standard learning-augmented model, the algorithm is supplied with some predictions about the future, and we seek to bound the competitive ratio of the algorithm in terms of some measure of the "prediction error", so that when the error is low the algorithm is near-optimal, and when the error is high the algorithm's performance gracefully decays. In the new model, the algorithm is again supplied with some predictions about the future, but we seek to bound the competitive ratio of the algorithm in terms of the probability that the prediction is correct. The goal is that even when the probability of correctness is a small constant, the algorithm still outperforms un-augmented algorithms.

For caching, the predictions are of the page in the cache which will take the longest time to be requested again (as in prior work on learning-augmented caching). The assumption is that with probability epsilon the prediction is exactly correct, and otherwise the prediction is a uniformly random page from the cache. This paper provides a simple algorithm OneStrike which achieves performance O(min(1/eps, log k)) and a more complex algorithm TwoStrike which achieves performance O(min(log(1/eps), log k)). They also show that if eps = 1/(k log k) then TwoStrike is optimal, i.e. it's impossible to beat Omega(log k).

For online set cover, the predictions are as follows. Each time an element is given, the algorithm needs to pick a set which contains that element (each element has a cost), and the prediction is supposed to supply one of the sets in the optimal solution which contains that element. The assumption is that with probability epsilon the prediction is in fact in the optimal solution. This paper provides a simple algorithm Set-Hedge which achieves competitive ratio O(1/eps), and shows that this is optimal. Finally, this algorithm is generalized to a class of "submodular-cost coverage problems".

**Questions:**

1) Is there a more comprehensive lower bound for caching? (e.g. for larger epsilon)
2) What happens if the "bad" predictions are allowed to be arbitrary? Is it possible that log(1/eps) is still attainable, or is there a stronger lower bound in this model?
3) In the set cover model, it's rather restrictive that the "good" prediction is required to be in the optimal solution. Does the proof generalize if we instead assume that the prediction is in a near-optimal solution? (e.g. a solution with cost within a constant factor of optimal).
4) More fundamentally, what if there are multiple optimal solutions and the "good" predictions at different steps are contained in possibly-different optimal solutions? It seems to me that the potential function proof might require it to be the same OPT at each step. Is there a reason to expect in practice that the prediction would be "consistent", i.e. drawn from a single optimal solution?

**Limitations:**

Yes

**Strengths And Weaknesses:**

In brief, this paper is original (introducing an interesting and practical twist to the traditional model of learning-augmented online algorithms) and clearly written. I checked correctness of the simpler caching algorithm (OneStrike), the online set covering algorithm (Set-Hedge), and the lower bounds. Set-Hedge is particularly elegant, whereas the algorithm/analysis of Two-Strike is rather complicated and ad-hoc (although this is a common issue in this literature). There are some weaknesses to the caching model/results which I expand on below. On the whole I think the conceptual contribution of this paper is interesting; on the other hand, there are significant ways in which the model/results are not entirely practically justified / could be strengthened (although these could reasonably be left to future work).

Strengths:
- The main strength of this paper is the new model. While it does not generalize prior work (it's incomparable), it introduces a new idea: it allows for some predictions to be arbitrarily erroneous, which may easily arise in practice when predictions are produced via machine learning.

Weaknesses:
- The weakness of this model (for both caching and set-cover) is that it requires the "good" predictions to be exactly correct. For caching, it seems often unreasonable to ask that the machine-learned model predicts which exact page will be requested farthest in the future with constant probability, since there may be a long tail of e.g. O(sqrt k) very rare pages in the cache which cannot be plausibly disambiguated. In such a setting we would expect that epsilon is roughly O(1/sqrt k), in which case (in this model) we do not improve upon the un-augmented O(log k) competitive ratio. In contrast, in the prediction error model, it's fine if all predictions have some noise. To put some numbers, if a phase has length O(k), then these hypothetical sqrt k rare pages would contribute at most roughly eta := k^{1.5} to the l1 error per phase; if OPT has O(k) cache misses in this phase, then the relative error eta/OPT is sqrt k, in which case we might expect (e.g. via [Wei20]) a competitive ratio of O(1). While this is a very rough back-of-the-envelope calculation, it suggests that either the augmented caching model introduced in this paper doesn't quite capture the strengths of machine-learned predictions, or the algorithm is not actually optimal.

- If I'm not missing some simple reduction, the caching lower bound of Omega(log 1/eps) claimed in this paper (Theorem B.6) is a little oversold. In particular, it is only proven for eps = 1/(k log k): that is, when the probability of getting the correct prediction is *lower-order* than the probability of getting any fixed page in the cache from a random prediction. This leaves open the possibility that for larger epsilon, the O(log 1/epsilon) algorithm can be improved (e.g. perhaps if epsilon = 1/sqrt(k) there is actually an algorithm with o(log k) competitive ratio). Indeed, this sort of phenomenon does happen in the prediction error model (parametrizing the competitive ratio by the relative error (l1 error)/OPT) [Wei20].

- Theorem 2.4 (analyzing the main caching algorithm TwoStrike) relies on a much stronger assumption than claimed in the abstract ("with predictions that are accurate with a small probability and arbitrarily inaccurate otherwise"). Specifically, with epsilon probability the prediction is assumed to be correct, and with the remaining probability the prediction is assumed to be a *uniformly random page from the cache*. This seems to me entirely unreasonable in a model which is motivated by machine-learned predictions. Errors made by machine-learning models tend to have severe correlations and biases, and are far from uniform.

---

> ### Author Response · Authors · 2022-08-02
> **Response to iZrF**
>
> > "The weakness of this model (for both caching and set-cover) is that it requires [...] the algorithm is not actually optimal."
>
> Indeed, our algorithm is more powerful. We do not really require suggestions to be precisely the FIF page; the algorithm works unchanged as long as the good suggestions tell us some page that is not requested again in the current phase. (The FIF page is just one such page.) So in the example above, any one of these $\sqrt{k}$ pages would count as a good suggestion for our purposes.
>
> We will add a discussion of this phenomenon, and more details about the requirements of the noisy oracle in the next version of the paper.
>
> > "If I'm not missing some simple reduction, the caching lower bound of Omega(log 1/eps) [...] by the relative error (l1 error)/OPT) [Wei20]."
>
> You are right: our lower bound only holds for the regime you mention. While we can probably get something for $\varepsilon \approx 1/k$, we do not have lower bounds for much higher values of $\varepsilon$, which is where our algorithms shine. We currently cannot rule out the possibility that there is an $o(\log k)$-competitive algorithm for $\varepsilon = 1/\sqrt{k}$; this remains a very exciting direction of research. We will change the language and make these issues clearer in the next version.
>
> > "Theorem 2.4 (analyzing the main caching algorithm TwoStrike) relies on [...] and are far from uniform."
>
> The use of uniformly random corruptions was to keep the model simple. Our algorithm works as long as the corrupted suggestions are drawn from a "diffuse" distribution over pages: namely, when each page is drawn with probability at most $O(1/\mathrm{poly}(\varepsilon)) \cdot 1/k$. (We will add the details to the final version: this follows by changes around line 590 of the paper.)
>
> That said, it seems difficult to remove all assumptions on the corrupted suggestions. Indeed, we have a very weak signal (we want correctness only with probability $\varepsilon$), so if the bad suggestions are  adversarial (and put a lot of the remaining $1-\varepsilon$ probability measure on a small set of pages which should not be evicted, since they are requested again in this phase), it seems difficult to beat an $O(1/\varepsilon)$ competitive ratio.
>
> We view both the uniform-random noise model and the diffuse noise model (which bounds the $\ell_\infty$ norm of the noise distribution at each step) as promising first steps in understanding predictions with noise, particularly in the range where the signal has very low magnitude ($\varepsilon \ll 1$). We hope that our work leads to further investigation of these kinds of oracles.
>
> Interestingly, independent work of Gamlath et al. [COLT 22] for the $k$-means clustering problem with noisy cluster-label predictions also considers very weak signals (correct with small probability), and for this setting they also consider uniformly random noise.
>
> > "Is there a more comprehensive lower bound for caching? (e.g. for larger epsilon)"
>
> No, please see response above.
>
> > "What happens if the "bad" predictions are allowed to be arbitrary? Is it possible that log(1/eps) is still attainable, or is there a stronger lower bound in this model?"
>
> No, please see response above.
>
> > "In the set cover model, it's rather restrictive that the "good" prediction is required to be in the optimal solution. Does the proof generalize if we instead assume that the prediction is in a near-optimal solution? (e.g. a solution with cost within a constant factor of optimal)."
>
> One possible source of confusion (which we will fix): what we call "$\mathrm{OPT}$" can actually be any reference/benchmark solution and not necessarily an optimal solution. We guarantee that we are comparable to the cost of this reference solution, up to the loss of $O(1/\varepsilon)$, so the interesting case is when this benchmark is itself near-optimal.
>
> > "More fundamentally, what if there are multiple optimal solutions [...] drawn from a single optimal solution?"
>
> We need some kind of consistency across the suggestions: each suggestion being independently drawn from *some* optimal solution is clearly not enough, else each suggestion may be completely arbitrary yet consistent with some completely different optimal solution. Our assumption should be viewed as saying that there is some solution of small cost consistent with all the "good" suggestions, which seems like a minimal assumption to make.

---

> > ### Comment · Reviewer_iZrF · 2022-08-06
> > **Response**
> >
> > Thanks for the reply.
> >
> > >  the algorithm works unchanged as long as the good suggestions tell us some page that is not requested again in the current phase
> >
> > It then seems to me that the prediction model does not really fully capture the strength of the algorithm.
> >
> > Also, something I do not recall being addressed in the paper - do prior algorithms (e.g. [Wei20]) perform poorly in this new prediction model? I think either empirical or theoretical evidence for this would substantially strengthen the paper.
> >
> > > Indeed, we have a very weak signal...it seems difficult to beat an $O(1/\epsilon)$ competitive ratio.
> >
> > Good point, this seems likely. It would be good to make this formal.
> >
> > > "OPT" can actually be any reference/benchmark solution and not necessarily an optimal solution.
> >
> > Ah, that makes sense.

---

> > > ### Author Response · Authors · 2022-08-09
> > > **Response to iZrF**
> > >
> > > > "Also, something I do not recall being addressed in the paper - do prior algorithms (e.g. [Wei20]) perform poorly in this new prediction model? I think either empirical or theoretical evidence for this would substantially strengthen the paper."
> > >
> > > Thanks for the suggestion. Indeed, we can show that Wei’s algorithm performs poorly in this prediction model --- it combines together Randomized Marking and BlindOracle (which in our terminology is called OneStrike). Since the BlindOracle algorithm just evicts the page suggested by the oracle, we take $n=k+1$ pages and construct a sequence of phases. In phase $i$ we request all pages except page $i$ round-robin, and do this $k$ times. The optimal strategy is to evict page $i$ at the start of this phase. But the algorithm follows the oracle blindly, so it will evict random pages due to bad suggestions $(1/\epsilon)$ times in expectation before getting a good suggestion and evicting page $i$. This happens in each phase, giving an expected cost $\Omega(1/\epsilon)$ times the optimum. Since Wei combines this with Randomized Marking which has an $\Omega(\log k)$ lower bound, by setting $\epsilon = 1/\log k$ and interleaving phases of the above lower bound sequence with phases of the lower bound for Randomized Marking, we get a sequence that causes Wei's algorithm to pay $\Omega(\log k)$ times $\mathrm{OPT}$, whereas our algorithm pays $O(\log(1/\epsilon) = O(\log \log k)$ times $\mathrm{OPT}$.
> > >
> > > If this sketch is not clear enough, we are happy to give further details. And we will add this example to the next version of the paper.

---

> > > > ### Comment · Reviewer_iZrF · 2022-08-09
> > > > **Thanks**
> > > >
> > > > Ah, that makes sense, thanks for answering my questions. I've updated my score.

---

### Official Review · Reviewer_jWJY · 2022-07-08

**Rating:** 6
**Confidence:** 4
**Soundness:** 3 good
**Presentation:** 4 excellent
**Contribution:** 4 excellent

**Summary:**

This paper considers the recently popular learning-augmented model in online algorithmic design. The authors use the minimum probability guarantee that a prediction is 100\% accurate to measure the quality of the prediction: say a prediction is ``$\epsilon$-accurate" if it is perfect with probability at least $1-\epsilon$. The paper presents several online algorithms which incorporate $\epsilon$-accurate predictions. For caching, a competitive ratio of $O(\log (1/\epsilon))$ is obtained, while for covering problems, a $O(1/\epsilon)$-competitive algorithm is given. The competitive lower bounds of caching and covering with $\epsilon$-accurate predictions are also discussed in the paper to show the optimality of the proposed algorithms.

**Questions:**

- In the proof of Lemma 2.9 (line 591-593), why the expected contribution of $p$ to BAD-STRIKES is at most $\sum_{i\geq 1} i {N \choose i} \frac{1}{k^{2i}}$? It is not clear to me where the term $\frac{1}{k^{2i}}$ comes from. Intuitively, this term should have a dependence on the accuracy parameter $\epsilon$.

**Limitations:**

I didn't see any potential negative societal impact.

**Strengths And Weaknesses:**

Strength:
- A novelty of this paper is using the accuracy guarantee instead of the traditional prediction error to measure the prediction quality. Although the accurate parameter $\epsilon$ can still be viewed as a particular prediction error, I still think this measure is interesting and could influence future work, where it is not clear that how to define a prediction error.

- The paper is technical. It proposes learning-augmented algorithms for classical online problems: caching and covering problems. Both algorithms are proved to have nearly optimal dependence of the accurate parameter $\epsilon$.

Weakness:
- There is no discussion about the learnability of such $\epsilon$-accurate predictions. Can we really learn them from historical data? Intuitively, it seems hard to get a 100\% accurate prediction with high probability.

- Experiments may be needed. Caching and covering problems are well studied in the learning-augmented model. The paper builds on a new prediction quality measure to give algorithms. So it might be better to conduct experiments to compare against previous learning-augmented algorithms.

- Line 76: ``the minimize the number of page swaps ".

- Line 84: ``FIF"  should be given the full name the first time it is used.

---

> ### Author Response · Authors · 2022-08-02
> **Response to jWJY**
>
> Thanks for your comments and suggestions; we will fix the typos mentioned above.
>
> > "There is no discussion about the learnability of such $\varepsilon$-accurate predictions. Can we really learn them from historical data? Intuitively, it seems hard to get a 100% accurate prediction with high probability."
>
> We agree that getting a 100% accurate prediction with high probability is unreasonable. Hence our focus is on the case when the probability $\varepsilon$ of getting a good prediction is small, so that the signal is very weak. The typical case we have in mind is when $\varepsilon = 0.01$, so that we get a good prediction only 1% of the time (and bad predictions 99% of the time). We still want to do better than the worst case. We feel this model is an interesting complement to most other prediction models, which require the predictions to be close (in some distance metric) to a good solution over the entire set of predictions.
>
> > "Experiments may be needed. Caching and covering problems are well studied in the learning-augmented model. The paper builds on a new prediction quality measure to give algorithms. So it might be better to conduct experiments to compare against previous learning-augmented algorithms."
>
> We are not aware of experimental evaluation in prior work on online covering with ML predictions (Bamas, Maggiori, and Svensson [NeurIPS 20] and Anand, Ge, Kumar, and Panigrahi [ICML 22]). In online caching with ML predictions, the only experimental results we are aware of are in the paper of Lykouris and Vassilvitskii [ICML 18] (the follow up works of Rohatgi [SODA] and Wei [APPROX] do not have experimental evaluation). It would be interesting to experimentally compare our algorithm with that of Lykouris and Vassilvitskii for the same data sets. That said, it is important to remember that the two algorithms are for different regimes in terms of noise: we require a very weak signal, while the Lykouris-Vassilvitskii algorithm is parameterized in terms of average $\ell_p$ error.
>
> > "In the proof of Lemma 2.9 (line 591-593), why the expected contribution of $p$ to BAD-STRIKES is at most $\sum_{i\geq 1}i{N \choose i}\frac{1}{k^{2i}}$? It is not clear to me where the term $\frac{1}{k^{2i}}$ comes from. Intuitively, this term should have a dependence on the accuracy parameter $\varepsilon$."
>
> The contribution of a non-FIF page $p$ to BadStrikes is essentially a binomial random variable, with the difference that two successes (rather than one) are needed to increment the BadStrikes counter. Hence, for page $p$ to contribute $i$ times to BadStrikes within the $N$ active time steps, it must be incorrectly predicted $2i$ times as the FIF within the $N$ active time steps. Now each non-FIF page $p$ (when in the cache) is incorrectly predicted with probability $(1-\varepsilon)\cdot 1/k \leq 1/k$, which gives $(1/k)^{2i}$. You are correct that there is a dependence on $\varepsilon$, but we drop this dependence on $\varepsilon$ and use the upper bound of $1/k$ for simplicity. (This approximation is always valid; moreover, since $\varepsilon$ is small, we lose little in making this approximation.)

---

> > ### Comment · Reviewer_iZrF · 2022-08-04
> > **Experimental results for caching**
> >
> > >  In online caching with ML predictions, the only experimental results we are aware of...
> >
> > There is also the following experimental paper which implements a number of the recent theoretical papers (of course, in the original prediction framework):
> >
> > Chłędowski, Jakub, Adam Polak, Bartosz Szabucki, and Konrad Tomasz Żołna. "Robust learning-augmented caching: An experimental study." In International Conference on Machine Learning, pp. 1920-1930. PMLR, 2021.

---

> > > ### Author Response · Authors · 2022-08-09
> > > **Response to iZrF**
> > >
> > > Thanks for the reference. Indeed this paper provides a good benchmark for comparing our algorithm with previous ones (at least with the ones using Belady's oracle). Moreover, their predictor can be used to test what values of $\epsilon$-accuracy can be obtained with the Transformer architecture. Doing a thorough experimental analysis of our framework and algorithms is a clear direction for future research, and we will include this in the conclusion section of the final version of our paper.

---

### Official Review · Reviewer_QMX1 · 2022-07-12

**Rating:** 5
**Confidence:** 4
**Soundness:** 4 excellent
**Presentation:** 3 good
**Contribution:** 2 fair

**Summary:**

This paper proposes and studies a model of learning-augmented online algorithms where future predictions are “$\varepsilon$-accurate”, but do not necessarily have low “error”. (An $\varepsilon$-accurate prediction is an accurate prediction that has been corrupted with probability $1-\varepsilon$.) The authors show $\varepsilon$-accuracy is sufficient to obtain beyond-worst-case performance in two instantiations of this model for online paging and online covering, respectively. This work fits into a growing literature on algorithm design, motivated by the increased availability of data and the structure inherent in many real-world problems, in which the input is augmented with “predictions” about the task at hand.

**Questions:**

- For the section on set cover, the predictions appear to be reliant on the choice of a distinguished solution $\mathsf{OPT}$ if there are multiple optimal solutions. Is there a way to define the model without reference to a specific $\mathsf{OPT}$ so that the results still hold?
- For the section on covering-type problems, it is great to see that the result applies in such generality. For such settings, are there empirical datasets where such a predictor—of which constraint to make binding—makes sense?

**Limitations:**

The authors discuss the limitations of their theoretical results in Section 4.

**Strengths And Weaknesses:**

- Strengths:
    - I appreciate the thought put towards considering a new model of prediction, as I find that consideration of models of prediction—while being a central driver of the results in this literature—has not received as much attention it deserves.
    - The technical results appear correct (although I did not verify all the details in the supplement); and the analyses are clean.
- Weaknesses:
    - While I like the idea of considering $\varepsilon$-accurate predictions (as it can encompass, e.g., heavy-tailed noise), I would have found the main result in the section on online paging more compelling if the analysis did not rely on uniformly random corruptions.
    - The $O(1/\varepsilon)$ bounds feel somewhat limited, as they should be somewhat expected when on a $\varepsilon$-fraction of the rounds the augmented hint tells the algorithm exactly the right action to take.
    - The paper would be stronger with a more thorough discussion (or perhaps empirical support) of the modeling assumptions and why they were chosen.
- Minor:
    - Line 84: The acronym “FIF” (assuming it means “furthest in the future”) should be defined before its first use, perhaps in Line 80.
    - Line 323: “revealted” → “revealed”
    - References: Reference [25] is a duplicate of [24]; likewise for [28] and [27].

---

> ### Author Response · Authors · 2022-08-02
> **Response to QMX1**
>
> > "While I like the idea [...] not rely on uniformly random corruptions."
>
> The use of uniformly random corruptions was to keep the model simple. Our algorithm works as long as the corrupted suggestions are drawn from a "diffuse" distribution over pages: namely, when each page is drawn with probability at most $O(1/\mathrm{poly}(\varepsilon)) \cdot 1/k$. (We will add the details to the final version: this follows by changes around line 590 of the paper.)
>
> That said, it seems difficult to remove all assumptions on the corrupted suggestions. Indeed, we have a very weak signal (we want correctness only with probability $\varepsilon$), so if the bad suggestions are  adversarial (and put a lot of the remaining $1-\varepsilon$ probability measure on a small set of pages which should not be evicted, since they are requested again in this phase), it seems difficult to beat an $O(1/\varepsilon)$ competitive ratio.
>
> We view both the uniform-random noise model and the diffuse noise model (which bounds the $\ell_\infty$ norm of the noise distribution at each step) as promising first steps in understanding predictions with noise, particularly in the range where the signal has very low magnitude ($\varepsilon \ll 1$). We hope that our work leads to further investigation of these kinds of oracles.
>
> Interestingly, independent work of Gamlath et al. [COLT 22] for the $k$-means clustering problem with noisy cluster-label predictions also considers very weak signals (correct with small probability), and for this setting they also consider uniformly random noise.
>
> > "The $O(1/\varepsilon)$ bounds feel somewhat limited..."
>
> While the $O(1/\varepsilon)$ bounds are natural, they do require care, since we need to show that the decisions we make during the other $1-\varepsilon$ fraction of the time do not cause the performance to degrade.
>
> The main difficulty is that our algorithms are unaware of "which" suggestions are correct. So, it's not just the case that the signal is weak (correct only $\varepsilon$ fraction of the time), but that this weak signal is mixed with large amounts of noise. Contrast this to a model where a suggestion is provided with probability $\varepsilon$, but when provided it is correct. In this latter model, a $1/\varepsilon$ bound is more straightforward and holds for a broader range of problems. (E.g., for problems like $k$-server we still don't know how to prove $O(1/\varepsilon)$-bounds in our model, but it is easy to show such a bound in the latter weaker model.)
>
> > "The paper would be stronger with a more thorough discussion (or perhaps empirical support) of the modeling assumptions and why they were chosen."
>
> We will definitely add a more detailed discussion to the final version. Thanks for your questions: they have helped bring the relevant modeling issues into sharper focus. We will also address the minor fixes you suggest.
>
> > "For the section on set cover, the predictions appear [...]"
>
> We need some kind of consistency across the suggestions: each suggestion being independently drawn from *some* optimal solution is clearly not enough, else each suggestion may be completely arbitrary yet consistent with some completely different optimal solution. Our assumption should be viewed as saying that there is some solution of small cost consistent with all the "good" suggestions, which seems like a minimal assumption to make.
>
> One possible source of confusion (which we will fix): what we call "$\mathrm{OPT}$" can actually be any reference/benchmark solution and not necessarily an optimal solution. We guarantee that we are comparable to the cost of this reference solution, up to the loss of $O(1/\varepsilon)$, so the interesting case is when this benchmark is itself near-optimal.
>
> > "For the section on covering-type problems, it is great to see that the result applies in such generality. For such settings, are there empirical datasets where such a predictor—of which constraint to make binding—makes sense?"
>
> The standard assumption in online covering problems is that all the covering constraints are binding. This is the case in our paper, as also in prior work on online covering with ML predictions such as Bamas, Maggiori, and Svensson [NeurIPS 20] and Anand, Ge, Kumar, and Panigrahi [ICML 22]. That said, if one wanted to incorporate the idea of non-binding constraints, one way of doing so would be to use a non-negative penalty function where each violated constraint incurs a given penalty. So, binding constraints have infinite penalty, while non-binding constraints have finite penalty (and the magnitude of the penalty allows one to express how binding different constraints are relative to one another).
>
> Regarding empirical data sets, unfortunately, we are not aware of any empirical results on online covering with ML predictions. The prior papers on this due to Bamas, Maggiori, and Svensson [NeurIPS 20] and Anand, Ge, Kumar, and Panigrahi [ICML 22] do not have experiments on online covering.

---

### Meta-Review · Area_Chair_arvv · 2022-08-24

**Recommendation:** Accept
**Confidence:** Less certain

**Metareview:**

This paper is part of a recent line of work that augments algorithms with ML predictions. It introduces an interesting new model where the quality of a prediction is measured by its accuracy, instead of its error, and gives strong competitive ratios for the online problems of caching and covering augmented with such predictions. This new model could influence future work in this area.

**Award:**

No

---

### Decision · Program_Chairs · 2022-09-14

Accept